# Theoretical Insights into Overparameterized Models in Multi-Task and Replay-Based Continual Learning

**Amin Banayeeanzade**                                                    *banayeea@usc.edu*
*Department of Computer Science*
*University of Southern California*

**Mahdi Soltanolkotabi**                                                  *soltanol@usc.edu*
*Department of Electrical and Computer Engineering*
*University of Southern California*

**Mohammad Rostami**                                                      *rostamim@usu.edu*
*Department of Computer Science*
*University of Southern California*

**Reviewed on OpenReview:** *https://openreview.net/forum?id=4zGPTOZwnU*

## Abstract

Multi-task learning (MTL) is a machine learning paradigm that aims to improve the generalization performance of a model on multiple related tasks by training it simultaneously on those tasks. Unlike MTL, where the model has instant access to the training data of all tasks, continual learning (CL) involves adapting to new sequentially arriving tasks over time without forgetting the previously acquired knowledge. Despite the wide practical adoption of CL and MTL and extensive literature on both areas, there remains a gap in the theoretical understanding of these methods when used with overparameterized models such as deep neural networks. This paper studies the overparameterized linear models as a proxy for more complex models. We develop theoretical results describing the effect of various system parameters on the model's performance in an MTL setup. Specifically, we study the impact of model size, dataset size, and task similarity on the generalization error and knowledge transfer. Additionally, we present theoretical results to characterize the performance of replay-based CL models. Our results reveal the impact of buffer size and model capacity on the forgetting rate in a CL setup and help shed light on some of the state-of-the-art CL methods. Finally, through extensive empirical evaluations, we demonstrate that our theoretical findings are also applicable to deep neural networks, offering valuable guidance for designing MTL and CL models in practice [1].

## 1 Introduction

The dominant machine learning paradigm involves training a single model on a single task and deploying the task-specific model for the targeted application. This approach, while effective for many applications, often results in a highly specialized model that lacks the ability to generalize across different yet related tasks. To overcome this issue, Multi-Task Learning (MTL) methods train a single model on multiple tasks simultaneously with the goal of benefiting from the similarities between a collection of related tasks to improve the efficiency of learning (Caruana, 1997). MTL not only enhances the performance of individual tasks but also facilitates the development of models that are more adaptable and capable of transferring knowledge from one task to another. This transferability is particularly valuable in real-world applications of deep learning where data can be scarce or imbalanced (Zhang & Yang, 2022).

---

[1]Code is available at https://github.com/aminbana/MTL-Theory .

In many real-world applications, the tasks are not available for offline training, and instead, the agent needs to learn new tasks sequentially as they are encountered. Learning in these settings is particularly challenging due to the phenomenon known as catastrophic forgetting, where learning new tasks can lead to a significant loss of previously acquired knowledge (French, 1999). Continual Learning (CL) is a solution to this challenge that builds on the foundation of MTL but with a focus on the model's ability to learn continuously over time without forgetting (Parisi et al., 2019).

Despite the practical success of integrating MTL and CL with deep learning, there still remains a critical need for a theoretical understanding of learning mechanisms in these methods (Crawshaw, 2020). Several efforts have been previously made to theoretically understand MTL (Baxter, 2000; Ben-David & Borbely, 2008). However, these endeavors are inapplicable to contemporary deep neural networks (DNN), where the models are heavily parameterized. In the overparameterized regime, DNNs exhibit peculiar generalization behaviors, where despite having more parameters than training samples, they can still generalize well to unseen test data (Zhang et al., 2017). An important unexplored area, however, is the effect of overparametrization on the generalizability of MTL models. More importantly, it is crucial to understand the impact of overparametrization on the possibility of effective cross-task knowledge transfer.

DNN models optimized with SGD inherit some specific behaviors of linear models, specifically in overparameterized regimes (Chizat et al., 2020). As a result, linear models have been widely studied as the first step toward understanding the deep double descent and benign overfitting (Hastie et al., 2022). However, previous investigations are limited to single-task learning paradigms with additive label noise. In this work, we provide theoretical characterizations of the linear overparameterized models in an MTL configuration for the first time. More specifically:

- We provide explicit expressions describing the expected generalization error and knowledge transfer of multi-task learners and compare them to single-task learners in a linear regression setup with i.i.d. Gaussian features and additive noise. Our results help to understand the system's behavior by highlighting the role and the interplay of various system parameters, including task similarity, model size, and sample size. For instance, we demonstrate that a peak exists in the generalization error curve of the multi-task learner when increasing model complexity, and the error peak stems not only from label noise but also from the dissimilarity across the tasks. We also highlight the effect of the sample size on the strength and location of the test error peak. Additionally, we measure knowledge transfer and indicate the conditions under which the tasks can be effectively learned together or interfere with each other.

- We provide similar results for continual learners and use them to explain the characteristics of replay-based CL methods. These results demonstrate the impact of replay buffer size on forgetting and the generalization error. Furthermore, we complete the theoretical results by analyzing the CL methods that use a combined strategy of relying on implicit regularization and rehearsal techniques. As a practical fruit, our results shed light on state-of-the-art replay-based CL methods by illustrating the effect of model size on reducing the effectiveness of memory buffer, especially in the presence of practical limitations where the replay buffer and model size can not be arbitrarily large.

- We empirically show that our findings for the linear models are generalizable to DNNs by conducting various experiments using different architectures and practical datasets such as CIFAR-100, Imagenet-R, and CUB-200. Our experiments show that the test error follows a similar trend observed in linear models, indicating that our theoretical results can help understand MTL in DNNs. We also provide experiments with replay-based CL methods and study the effect of memory buffer size in practical setups with DNNs.

## 2 Related Work

**Multi-Task Learning**  Several theoretical efforts have been made to understand the benefits of multi-task learning. In one branch, the statistical learning theory (SLT) toolkit was used to derive generalization bounds for MTL models. These efforts include using VC dimension (Baxter, 2000), covering number (Pentina & Ben-David, 2015), Rademacher complexity (Yousefi et al., 2018), and algorithmic stability (Zhang, 2015). This line of work mostly focuses on defining notions of task similarity in the SLT sense (Ben-David & Borbely, 2008) and deriving generalization bounds as a function of the number of tasks, training set size, and task

similarity (Zhang & Yang, 2022). However, these methods are inapplicable to the contemporary paradigm of overparametrized DNNs (Allen-Zhu et al., 2019), mainly because these models achieve zero training error and perfectly memorize the training data, yet they still generalize well to the unseen test set.

Additionally, some previous work studied multi-variate regression models, but their analysis or setup fundamentally differs from ours. For instance, (Lounici et al., 2009) study sparse systems of equations with the goal of union support recovery (Kolar et al., 2011; Obozinski et al., 2011). More relevant to our work, Wu et al. (2020) analyzes a two-layer architecture with a shared module across tasks and separate output modules for each task. They derive an upper bound on the performance of an optimal multi-task learning (MTL) model in a two-task scenario, assuming data abundance and finite model width, and examine the impact of task alignment on MTL performance. In contrast to our work, these studies do not characterize the exact convergence point of SGD in the MTL setup or explicitly derive the system's generalization performance in the overparameterized regime.

**Continual Learning** CL methods predominantly use either model regularization (Kirkpatrick et al., 2017) or experience replay techniques (Schaul et al., 2015; Rolnick et al., 2019) to tackle catastrophic forgetting. Model regularization is based on consolidating the model parameters by penalizing drifts in the parameter space during model updates with the goal of preserving the older knowledge (Kirkpatrick et al., 2017; Zenke et al., 2017; Aljundi et al., 2018). The core idea for experience replay is to keep a portion of the training set in a separate memory buffer (Schaul et al., 2015). These samples are representative of previously encountered distributions and are replayed back during learning subsequent tasks along with the current task data. The stored samples enable the model to revisit and learn from the distributions it has encountered before, facilitating the retention of acquired knowledge from past tasks (Rolnick et al., 2019; Huang et al., 2024; Zhou et al., 2023). Memory-based methods have proven to be highly effective, and their variations often lead to the optimal approach in practice (Parisi et al., 2019).

Several theoretical papers exist on understanding continual learning models from different aspects (Doan et al., 2021; Yin et al., 2021; Chen et al., 2022). The most relevant to our work examines linear CL models under a Gaussian data model when trained sequentially using SGD as the data for new tasks arrive. For instance, Goldfarb & Hand (2023) derives an upper bound for the generalization error of such models, and Lin et al. (2023) refines this result by proposing a closed-form characterization of the error. Furthermore, Goldfarb et al. (2024) studies the interplay between forgetting and task similarity as a function of different levels of overparametrization under a more general data model. More recently, Guha & Lakshman (2024) extends the theoretical investigation to feed-forward neural networks to show that increasing network widths to reduce forgetting yields diminishing returns. Although these results are valuable and generally comply with our findings, they fall into the regularization CL-based methods. This is due to the fact that they merely rely on the implicit bias of SGD as a regularizer that tends to the solutions closer to the weights of the previous task, while in our study, we equip the continual learner with an external memory buffer and study the closed form characteristics of the error as a function of different system parameters, especially the memory size. Moreover, some of the mentioned results (e.g., Lin et al. (2023)) can be seen as a special case of our studies in Section 4 when memory size is zero.

**Overparameterization and Double Descent** In classical learning theory, it was widely accepted that increasing a model's complexity would initially reduce test error, but beyond a certain point, the error would rise again due to overfitting. However, the empirically observed double descent curve reveals that after the model complexity surpasses a certain threshold, making the model overparameterized, the test error surprisingly starts to decrease again (Nakkiran et al., 2021). In fact, state-of-the-art DNNs operate in the overparameterized regime, meaning that the model can perfectly fit the training data and achieve near-zero training errors while paradoxically still being able to generalize well to new unseen data, a phenomenon known as benign overfitting (Zhang et al., 2017; Cao et al., 2022).

Double descent is not specific to DNNs (Belkin et al., 2019). For instance, linear regression exhibits similar behaviors under certain assumptions (Belkin et al., 2020). On the other hand, recent literature has pointed out a direct connection between linear models and more complex models such as neural networks optimized with SGD (Jacot et al., 2018; Gunasekar et al., 2018; Oymak & Soltanolkotabi, 2019). Consequently, several studies investigated linear models as a proxy for more complex models such as DNNs (Hastie et al., 2022).

However, these works are primarily focused on single-task setups and are not capable of capturing task notions in MTL setups. In contrast, our work analyzes the MTL case where multiple tasks are learned together. Considering that even a simple multi-label classification problem is a special case of MTL, our work is essential in understanding commonly used overparameterized DNN models.

## 3 Theoretical Results on MTL in Overparameterized Regimes

We first define the problem setting and then offer our theoretical results.

### 3.1 Problem Formulation

Consider a set of $T$ learning tasks with a linear ground truth model for each task. Specifically, for task $t$, assume an input feature vector $\bar{x}_t \in \mathbb{R}^{s_t}$ and an output $y \in \mathbb{R}$ given by

$$y = \bar{x}_t^\top \bar{w}_t^* + z, \tag{1}$$

where $\bar{w}_t^* \in \mathbb{R}^{s_t}$ denotes the optimal parameters, and $z \in \mathbb{R}$ is the random noise (Lin et al., 2023; Belkin et al., 2018). The specific true feature space for each task is unknown, and the features for a specific task may not be useful to other tasks. Since we seek to use a single MTL model to learn all tasks, we consider a larger parameter space with size $p$ that encompasses all true features.

More formally, consider a global set of features indexed by $1, 2, \cdots$. We assume that the true set of features for task $t$ is denoted by $S_t$ such that $|S_t| = s_t$. Among all, we choose a set of $p$ features denoted by $\mathcal{W}$ to train our model, assuming that $\cup_{t=1}^T S_t \subseteq \mathcal{W}$. With this in mind, we define the expanded ground truth for task $t$ by introducing $w_t^* \in \mathbb{R}^p$, where $w_t^*$ is the same as $\bar{w}_t^*$ in the $S_t$ indices and filled by zero in the remaining $p - s_t$ places (Lin et al., 2023).

**Data Model.** We consider a training dataset of size $n_t$ for task $t$ represented as $\mathcal{D}_t = \{(x_{t,i}, y_{t,i})\}_{i=1}^{n_t}$, where $x_{t,i} \in \mathbb{R}^p$ and $y_{t,i} = x_{t,i}^\top w_t^* + z_{t,i}$. We also assume the features and noise are i.i.d. according to the following distributions:

$$x_{t,i} \sim \mathcal{N}(0, I_p) \quad \text{and} \quad z_{t,i} \sim \mathcal{N}(0, \sigma^2). \tag{2}$$

Here, $\sigma^2$ denotes the noise strength.

**Data Model in Matrix Form.** Consider a matrix representation by stacking the training data as $X_t := [x_{t,1}, x_{t,2}, ..., x_{t,n_t}] \in \mathbb{R}^{p \times n_t}$, $y_t := [y_{t,1}, y_{t,2}, ..., y_{t,n_t}]^\top$, and the noise vector as $z_t = [z_{t,1}, z_{t,2}, ..., z_{t,n_t}]^\top$, to summarize the data generation process as

$$y_t = X_t^\top w_t^* + z_t. \tag{3}$$

Similarly, we might stack the training data of all tasks to build $X_{1:T} \in \mathbb{R}^{p \times \bar{n}}$, $y_{1:T} \in \mathbb{R}^{\bar{n}}$ and $z_{1:T} \in \mathbb{R}^{\bar{n}}$ where $\bar{n} = \sum_{t=1}^T n_t$ is the total number of training samples.

**Single-Task Learning (STL)** To train a single-task learner for task $t$, we consider the standard setting in which the mean-squared-error loss function on $\mathcal{D}_t$ is optimized:

$$w_t = \operatorname*{arg\,min}_{w \in \mathbb{R}^p} \frac{1}{n_t} \|X_t^\top w - y_t\|_2^2. \tag{4}$$

In the underparameterized regime where $p < n_t$, there is a unique solution to minimizing the loss, given by $w_t = (X_t X_t^\top)^{-1} X_t y_t$. In contrast, in the overparameterized regime where $p > n_t$, there are infinite solutions with zero training error. Among all solutions, we are particularly interested in the solution with minimum

$\ell_2$-norm, since it is the corresponding convergence point of applying stochastic gradient descent (SGD) on Equation 4 (Gunasekar et al., 2018) when starting from zero initialization. In fact, $w_t$ in this case is obtained by equivalently solving the following constrained optimization:

$$w_t = \underset{w \in \mathbb{R}^p}{\arg\min} \|w\|_2^2 \quad \text{s.t.} \quad X_t^\top w = y_t. \tag{5}$$

Since we are interested in the overparameterized regime, our focus is on solving Equation 5.

**Single-Task Generalization Error.** To evaluate the generalization performance of $w$ on task $t$, we use the following test loss:

$$\mathcal{L}_t(w) = \mathbb{E}_{x,z}[(x^\top w - y)^2] = \|w - w_t^*\|_2^2 + \sigma^2. \tag{6}$$

In what follows, we drop $\sigma^2$ and only use $\mathcal{L}_t(w) = \|w - w_t^*\|_2^2$ as a comparison criterion. As a prelude to our work, we review the following theoretical results on the generalization error of single-task learners:

**Theorem 3.1.** *(Hastie et al. (2022)) When $n_t \geq p + 2$, the single-task learner described in Equation 4 achieves*

$$\mathbb{E}[\mathcal{L}_t(w_t)] = \frac{p\sigma^2}{n_t - p - 1}, \tag{7}$$

*and when $p \geq n_t + 2$, the single-task learner described in Equation 5 obtains*

$$\mathbb{E}[\mathcal{L}_t(w_t)] = (1 - \frac{n_t}{p})\|w_t^*\|_2^2 + \frac{n_t\sigma^2}{p - n_t - 1}, \tag{8}$$

*where the expectation is due to randomness of $X_t$ and $z_t$.*

This result characterizes the generalization error of single-task learning (STL) across both the underparameterized and overparameterized regimes. To better understand this theorem, fix the number of training samples $n$ and analyze how the error evolves as a function of the number of parameters $p$. In the underparameterized regime ($p < n$) the learner doesn't have enough capacity to fit all training samples. This serves as a regularization and leads to improved generalization, as the learned parameters approximate the optimal solution with low bias but some variance. As $p$ increases in this regime, the variance grows, leading to an overall increase in error until the interpolation threshold ($p \approx n$) is reached. At this critical point, there exists exactly one predictor that can perfectly fit the training data. However, this solution is highly sensitive to noise, as small fluctuations in the training data lead to large variations in the learned parameters, resulting in a sharp spike in test error.

Beyond the interpolation threshold, as $p$ continues to increase, the model enters the overparameterized regime, where it can completely memorize the training data, driving the training error to zero. Despite this apparent overfitting, the model exhibits benign overfitting, where increasing $p$ paradoxically reduces the influence of noise ($\frac{n_t\sigma^2}{p - n_t - 1} \to 0$ as $p \to \infty$). As discussed in Section 2, this surprising behavior is associated with the *implicit bias of SGD*, which biases the optimization toward low-norm solutions. With this in mind, we are ready to present our results on multi-task learning.

### 3.2 Main Results on Multi-Task Learning (MTL)

In this section, we present our main results for MTL and the deductions our results imply. Consider a multi-task learner that simultaneously learns all tasks by solving the optimization given in Equation 5 using the training data of all tasks:

$$w_{1:T} = \underset{w \in \mathbb{R}^p}{\arg\min} \|w\|_2^2 \quad \text{s.t.} \quad X_{1:T}^\top w = y_{1:T}. \tag{9}$$

To evaluate model performance, we utilize two metrics:

- *Average generalization error* reflects the overall generalization of the model, averaged across all tasks, i.e.,

$$G(w_{1:T}) := \frac{1}{T} \sum_{t=1}^{T} \mathcal{L}_t(w_{1:T}). \tag{10}$$

- *Average knowledge transfer* is a metric to measure the gain of mutual cross-task knowledge transfer when comparing MTL to STL, i.e.,

$$K(w_{1:T}) := \frac{1}{T} \sum_{t=1}^{T} \left[ \mathcal{L}_t(w_t) - \mathcal{L}_t(w_{1:T}) \right]. \tag{11}$$

Notably, a lower generalization and a higher knowledge transfer are more desirable. In what comes next, we offer the main results of the paper on the generalization error and knowledge transfer of an overparameterized multi-task learner.

**Theorem 3.2.** *The multi-task learner described in Equation 9 in the overparameterized regime, where $p \geq \bar{n} + 2$, has the following exact generalization error and knowledge transfer:*

$$\mathbb{E}[G(w_{1:T})] = \underbrace{\sum_{t=1}^{T} \sum_{t'=1}^{T} \frac{n_{t'}}{Tp} (1 + \frac{Tn_t}{2(p - \bar{n} - 1)}) \|w_t^* - w_{t'}^*\|_2^2}_{term\ G_1} + \underbrace{\frac{1}{T}(1 - \frac{\bar{n}}{p}) \sum_{t=1}^{T} \|w_t^*\|_2^2}_{term\ G_2} + \underbrace{\frac{\bar{n}\sigma^2}{p - \bar{n} - 1}}_{term\ G_3}, \tag{12}$$

$$\mathbb{E}[K(w_{1:T})] = \underbrace{2 \sum_{t=1}^{T} \sum_{t'=1}^{T} \frac{n_{t'}}{Tp} (1 + \frac{Tn_t}{2(p - \bar{n} - 1)}) \langle w_t^*, w_{t'}^* \rangle}_{term\ K_1}$$
$$\underbrace{- (1 + \frac{1}{T} + \frac{\bar{n}}{p - \bar{n} - 1}) \sum_{t=1}^{T} \frac{n_t}{p} \|w_t^*\|_2^2}_{term\ K_2} \underbrace{- \frac{\bar{n}\sigma^2}{p - \bar{n} - 1} + \frac{1}{T} \sum_{t=1}^{T} \frac{n_t \sigma^2}{p - n_t - 1}}_{term\ K_3}, \tag{13}$$

where $\langle w_t^*, w_{t'}^* \rangle$ denotes the inner product of the two vectors and the expectation is due to the randomness of $X_{1:T}$ and $z_{1:T}$.

Proofs for all theorems are provided in Appendix A. To the best of our knowledge, this is the first theoretical analysis of multi-task learning in overparameterized linear models providing an exact closed-form expression under non-asymptotic conditions. Setting $T = 1$ recovers the prior STL results (Hastie et al., 2022), which means that our theorem is a more generalized version of Theorem 3.1.

Theorem 3.2 precisely characterizes various phenomena that occur in the multi-task setting. Terms $G_2$ and $G_3$, which also appear in the STL setup, correspond to the task norm and noise strength. However, term $G_1$, which is specific to the MTL configuration, appears due to the distance between the optimal task vectors and is directly affected by task similarities. In fact, this observation motivates us to define a notion of similarity between any two tasks $t$ and $t'$ as the Euclidean distance between the optimal task vectors, i.e. $\|w_t - w_{t'}\|_2$.

To achieve a better insight into the results of Theorem 3.2, Figures 1a and 1b provide a visualization of the empirical average generalization error for different values of $p$, $\sigma$, and task similarities, as well as those predicted by our theory. Concretely, we have the following observations:

1. **Interpolation threshold shifts**. Similar to the single-task learner, an interpolation threshold exists for the multi-task learner. For both models, the average error increases at the threshold and decreases afterward. However, the interpolation threshold for the single-task learner occurs at $p = n_t$, while for the multi-task learner, it happens at $p = \bar{n}$, which is the point where the number of MTL training samples matches the number of parameters.

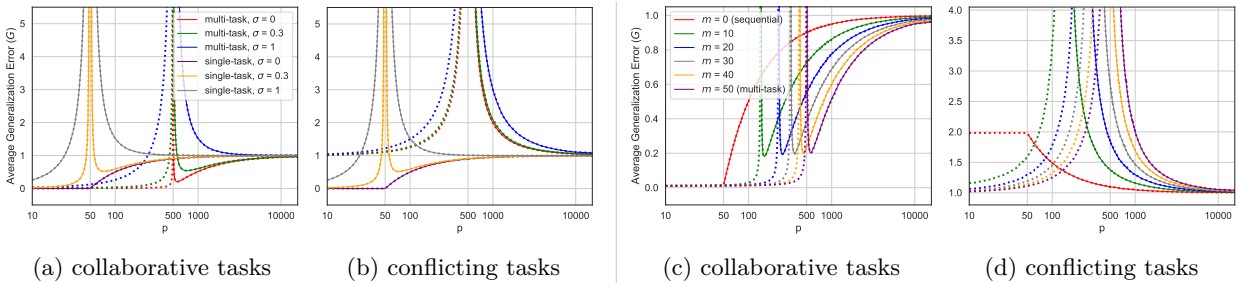

Figure 1: The average generalization error of multi-task, single-task and continual learners w.r.t. to the model size $p$. (a) and (b) compare the MTL and STL for different levels of noise strength $\sigma$. (c) and (d) present the generalization error of replay-based continual learners for different memory sizes $m$ with zero noise. For all plots, $T = 10$, and for all tasks $n_t = 50$ and $\|w_t^*\|^2 = 1$. In subfigures (a) and (c), the tasks are designed to be collaborative by adjusting $\langle w_t^*, w_{t'}^* \rangle = \cos \frac{\pi}{8}$ for every pair of task vectors. In subfigures (b) and (d), the tasks are chosen to be conflicting by setting $\langle w_t^*, w_{t'}^* \rangle = \cos \frac{7\pi}{8}$. The solid lines represent the results from theoretical predictions. The dot marks are the empirical evaluations averaged over 500 repetitions and are perfectly aligned with the theoretical results in the overparameterized regime.

2. **Not all tasks can be learned together**. The presence of the term $\|w_t^* - w_{t'}^*\|_2^2$ in $G_1$ indicates the impact of task similarity on the overall generalization performance. More specifically, if the tasks are highly similar, meaning that their optimal parameters are close, utilizing the training dataset of one task can improve the test generalization of the other task. However, when using a multi-task learner to simultaneously learn tasks with considerable gaps, interference happens in the parameters space and leads to poor performance.

   The knowledge transfer also tightly depends on task similarity. In fact, the sign of the $K_1$ term is closely correlated to the pairwise cosine similarity of the tasks. With conflicting tasks, negative knowledge transfer (Wang et al., 2019) happens and the multi-task learner underperforms the single-task learner. In addition to that, the $K_3$ term in Equation 13 reveals that with the same $p$, the multi-task learner performs worse against the noise. However, to fairly compare the models, notice that the multi-task learner, which has only $p$ parameters, is being compared against an average of $T$ single-task learners with $T \times p$ overall parameters.

   Finally, collaboration or conflict across the tasks is observable in models with limited capacity. In other words, when $p \to \infty$, both $G_1$ and $K_1$ terms vanish, meaning that neither collaboration nor interference happens anymore in infinite wide models.

3. **Task dissimilarity and noise intensify the error peak**. Recall that the STL test error peak that exists at the interpolation threshold appears because of the high variance of the learned predictor. This behavior stems only from the label noise and intensifies as $\sigma$ grows larger. Nevertheless, when $\sigma = 0$, meaning that there exists no label noise in the training set, no error peak happens at the interpolation threshold of the single-task learner. Meanwhile, in an MTL setup, there are two sources for the error peak. Not only can the label noise cause the peak through the term $G_3$, but task dissimilarity can also contribute through the term $G_1$, meaning that highly dissimilar tasks can produce larger error peaks. In fact, the stochasticity of data points (i.e., $X$), together with high task dissimilarity, can lead to highly variant predictors with poor generalization behavior. This observation explains earlier empirical findings where the double-descent phenomenon was observed even in the absence of label noise (Nakkiran et al., 2021) and also distinguishes the study of MTL models from STL learners.

4. **Descent floor exists when tasks are collaborative**. When tasks are collaborative and the noise is not too strong, increasing $p$ in the overparameterized regime can be a double-edged sword. On one hand, increasing the number of parameters can help by reducing the effect of noise through the term $G_3$. On the other hand, increasing $p$ to $\infty$ can kill the positive knowledge transfer across the tasks. There exists a point in the middle where the effect of both mechanisms matches and a descent floor appears in the generalization error (see Figure 1a as an example).

Although these observations are derived from our theoretical analysis of linear models, in Section 5, we will use an empirical approach to demonstrate that DNNs trained with SGD also exhibit similar behaviors. Consequently, our theoretical insights can extend to and enhance the understanding of more complex models.

## 4 Theoretical Results on CL in Overparameterized Regimes

Continual learning can be considered a special type of MTL, with an extra constraint that the tasks arrive sequentially during a continual episode. At each timestep $t$ the model is only exposed to $\mathcal{D}_t$, and the goal is to continually train a model that performs well across all tasks at the end of the episode. Note that the full data for past learned tasks is not accessible when the current task is being learned. In this case, sequentially fine-tuning on the most recent task at hand results in a model that performs well on the most recent task, but with degraded performance on previous tasks, a phenomenon which is called *catastrophic forgetting*. Formally, let $w_{\vec{t}}$ denote the weights of the continual learner at timestep $t$ after sequentially observing tasks from 1 to $t$. *Average forgetting* (lower is better) reflects the amount of negative backward knowledge transfer in a continual learner, defined as $F(w_{\vec{T}}) := \frac{1}{T-1} \sum_{t=1}^{T-1} [\mathcal{L}_t(w_{\vec{T}}) - \mathcal{L}_t(w_{\vec{t}})]$. For simplicity, in the rest of this section, we assume that $n_t = n$ for all tasks.

**Implicit Regularization for Continual Learning.** Prior to our work, recent theoretical CL studies relied on the implicit regularization of SGD to achieve a continual learning algorithm and analyzed its characteristics (Evron et al., 2022; Lin et al., 2023; Goldfarb & Hand, 2023; Goldfarb et al., 2024). In other words, they start from a zero initialization of weights, i.e. $w_{\vec{0}} = 0$, and simply use SGD to fit the training data of the new tasks as the tasks arrive sequentially. Because of the implicit bias of SGD, this method is equivalent to the following optimization:

$$w_{\overrightarrow{t+1}} = \underset{w \in \mathbb{R}^p}{\arg\min} \|w - w_{\vec{t}}\|_2^2 \quad \text{s.t.} \quad X_{t+1}^\top w = y_{t+1}, \tag{14}$$

where $w_{\vec{0}} = 0 \in \mathbb{R}^p$. This continual learner, often called *sequential finetuning* in CL, is not effective enough in practice and yields high forgetting. Even further, using explicit regularization can not solve the issue in practice and therefore, there is a need to study more useful CL strategies.

**Replay-Based Continual Learning.** Most successful methods in deep continual learning benefit from memory replay buffer (van de Ven et al., 2022). These methods store a small portion of the training data from each task. When learning new tasks, the stored samples are used to either regularize or retrain the model. In this section, we seek to theoretically study replay-based continual learning in the context of overparameterized linear models. Specifically, assume we have access to $m_i$ samples for task $i$ in the memory buffer when solving task $t + 1$. Let the memory size be $\bar{m}_t = \sum_{i=1}^{t} m_i$. We use the notation $\hat{X}_{1:t} \in \mathbb{R}^{p \times \bar{m}_t}$ and $\hat{y}_{1:t} \in \mathbb{R}^{\bar{m}_t}$ to denote the training data and their labels in the memory.

A simple strategy to update the weights, in this case, is to ignore prior weights learned so far, start again from the zero-initialized weights, and use SGD to fit both the data of the new task as well as the older data stored in the memory buffer. The described replay-based continual learning optimization is equivalent to

$$w_{\overrightarrow{t+1}} = \underset{w \in \mathbb{R}^p}{\arg\min} \|w\|_2^2 \quad \text{s.t.} \quad X_{t+1}^\top w = y_{t+1}, \quad \hat{X}_{1:t}^\top w = \hat{y}_{1:t}. \tag{15}$$

As opposed to Equation 14, the continual learner introduced here, tends toward low-norm solutions, but does not seek to minimize its distance with the weights acquired so far from previous tasks. Therefore, we name this strategy as pure replay-based continual learning.

**Implicit Regularization+Replay-Based Continual Learning.** Lastly, we can think of a continual learner that uses a mixed strategy of relying on both the replay memory buffer as well as the implicit regularization, i.e., to store older samples in the memory and also sequentially update the weights using SGD without reinitializing to zero. This strategy can be formulated as solving

$$w_{\overrightarrow{t+1}} = \underset{w \in \mathbb{R}^p}{\arg\min} \|w - w_{\vec{t}}\|_2^2 \quad \text{s.t.} \quad X_{t+1}^\top w = y_{t+1}, \quad \hat{X}_{1:t}^\top w = \hat{y}_{1:t}, \tag{16}$$

where $w_{\vec{0}} = 0 \in \mathbb{R}^p$. Notice that the optimization described in Equation 14 is a special case of Equation 16 when $m = 0$.

## 4.1 Theoretical Results on CL

In this section, we theoretically study the continual learners described in Equations 15 and 16. With a closer look, solving Equation 15 at $t = T$ is a specific case of multi-task learning described in Equation 9, achieved by setting $n_1 = m_1, ..., n_{T-1} = m_{T-1}$ and $n_T = n$. Therefore, we avoid repeating the theoretical results, and instead, we provide a corollary for the two-task case.

**Theorem 4.1.** *Assume $T = 2$, $\sigma = 0$, $m_1 = m$, $n_1 = n_2 = n$ and $\|w_1^*\| = \|w_2^*\|$. When $p \geq n + m + 2$, for the continual learner described in Equation 15, it holds that*

$$\mathbb{E}[G(w_{\vec{T}})] = \underbrace{\frac{n}{2p}(1 + \frac{m}{n} + \frac{2m}{p - (n + m) - 1})\|w_1^* - w_2^*\|_2^2}_{term\ G_1} + \underbrace{(1 - \frac{n + m}{p})\|w_1^*\|_2^2}_{term\ G_2}, \tag{17}$$

$$\mathbb{E}[F(w_{\vec{T}})] = \underbrace{\frac{n}{p}(1 + \frac{m}{p - (n + m) - 1})\|w_1^* - w_2^*\|_2^2}_{term\ F_1} - \underbrace{\frac{m}{p}\|w_1^*\|_2^2}_{term\ F_2}. \tag{18}$$

The generalization error of the regularization+replay-based learner is more complicated compared to the pure replay-based model and requires more delicate investigation. In fact, the dependence of $w_{\vec{t}}$ to the samples in the memory buffer is the source of such complication and produces many cross-terms in the final expression. To keep the results understandable and intuitive, we present the results for the two tasks case here and leave the full form for Appendix A.4.

**Theorem 4.2.** *Assume $T = 2$, $\sigma = 0$, $m_1 = m$, $n_1 = n_2 = n$ and $\|w_1^*\| = \|w_2^*\|$. When $p \geq n + m + 2$, for the continual learner described in Equation 16, it holds that*

$$\mathbb{E}[G(w_{\vec{T}})] = \underbrace{\frac{n}{2p}(2 - \frac{n - m}{p - m} + \frac{2m}{p - (n + m) - 1})\|w_1^* - w_2^*\|_2^2}_{term\ G_1} + \underbrace{(1 - \frac{n}{p - m})(1 - \frac{n}{p})\|w_1^*\|_2^2}_{term\ G_2}, \tag{19}$$

$$\mathbb{E}[F(w_{\vec{T}})] = \underbrace{\frac{n}{p}(1 + \frac{m}{p - (n + m) - 1})\|w_1^* - w_2^*\|_2^2}_{term\ F_1} - \underbrace{\frac{n}{p - m}(1 - \frac{n}{p})\|w_1^*\|_2^2}_{term\ F_2}. \tag{20}$$

Equations 17 to 20 reveal the effect of memory capacity. Setting $m = 0$ yields the results for the sequential learner described in Equation 14. Increasing the memory size reduces the forgetting and the generalization error through the terms $G_2$ and $F_2$, respectively. However, a larger memory size can be harmful through the terms $G_1$ and $F_1$ if the tasks are highly dissimilar. Additionally, the full MTL model is recovered when $m = n$, meaning that all samples from the previous tasks are stored in the buffer.

An interesting observation is the effect of model size on the forgetting and the generalization error. With a larger $p$, both the positive and negative terms in the forgetting vanish. This observation suggests that bigger models with more capacity are less vulnerable to forgetting (Goldfarb & Hand, 2023). Moreover, the next observation is that a large $p$ reduces the effect of memory size which means a natural trade-off exists in practical applications with limited physical memory. In other words, one may consider investing the hardware in deploying larger models or consider a larger memory buffer for training samples. In fact, this is an essential result that sheds light on several state-of-the-art continual learning models where they reported that sample memory is not enough and the best solution is to also keep some parts of the architecture in the memory (Zhou et al., 2023; Wang et al., 2022; Douillard et al., 2022). We revisit this result in Section 5 when studying DNNs.

Although we presented the $T = 2$ versions of the theorems for better comprehension, the mentioned phenomena are also observable in the $T > 2$ case as presented in Appendices A.3 and A.4. Figures 1c and 1d visualize the error behavior of the replay-based model for different values of $m$ when learning $T = 10$ tasks. As demonstrated, the location of the error peak and its strength are affected by memory size, $m$. Moreover, it is surprisingly observable that the continual learner can sometimes outperform the MTL baseline even when tasks are collaborative, especially at the multi-task learner's interpolation threshold.

## 5 Empirical Exploration for Deep Neural Networks

### 5.1 Empirical Results

Although fully understanding studying DNNs through theory seems infeasible, recent literature has pointed out connections between DNNs optimized by SGD and linear models and used these models as a proxy to study DNNs (Please refer to Appendix B for a discussion on the connection between overparameterized DNNs and linear models). We follow a similar approach and empirically demonstrate how our results can provide insights for understanding and designing deep MTL and CL models. In summary, our most important observations are as follows:

- MTL DNNs experience a test error peak at the interpolation threshold, but unlike STL models, the peak not only depends on the label noise but is also intensified when tasks are highly dissimilar or ineffective parameter sharing is happening (Figure 2), a consistent behavior across various practical datasets and architecture (Figures 4 and 5).

- The distance of optimal task-specific parameters correlates with the amount of conflict that is happening across different tasks and can provide evidence for deciding which layers to decouple (Figure 3).

- Memory-based continual learners also undergo a double descent as the number of parameters is increased. The characteristics of the error peak depend on the memory size as well as task-similarity and training size (Figure 6).

- High overparametrization can diminish the negative knowledge transfer (forgetting). Additionally, overparametrized models benefit less from explicit memory buffers. Therefore, it is important to adjust the resources accordingly between model parameters and raw training samples (Table 1).

### 5.2 Experimental Setup

Our experiments are performed using three datasets and three backbone architectures to offer the generalizability and scalability of our results to non-linear models. It is notable that all experiments are repeated at least 3 times, and the average values are reported. In the subsequent paragraphs, we present the most important details of our experiments and leave further information for Appendix C.

**Datasets.** We use CIFAR-100, ImageNet-R (IN-R), and CUB-200 datasets in our experiments. We generate MTL or CL tasks by randomly splitting these datasets into 10 tasks, with an equal number of classes in each task (van de Ven et al., 2022). No specific data augmentation or preprocessing techniques were applied, except for resizing to accommodate the input requirements of different backbones.

**Architecture.** We utilize the ResNet-18, ResNet-50 and ViT-B-16 as three different backbones in our experiments. ResNet-50 and ViT-B-16 were respectively pretrained on Imagenet-1k and Imagenet-21k and used as frozen feature extractors shared across all tasks. However, we used two versions of ResNet-18, namely, a frozen version, pretrained on CIFAR-10 and an unfrozen version with random initialization.

Our main focus is on the models operating in overparameterized regimes. Therefore, we leverage models that are complex enough to ensure that the training data is completely overfitted. When using the pretrained backbones, we implement a shared 3-layer MLP on top of the feature extractor followed by a classification head. We use the width of the MLP to manipulate the model's complexity by changing the number of hidden neurons for all of the 3 layers. On the other hand, when the backbone is trainable, we adjust the

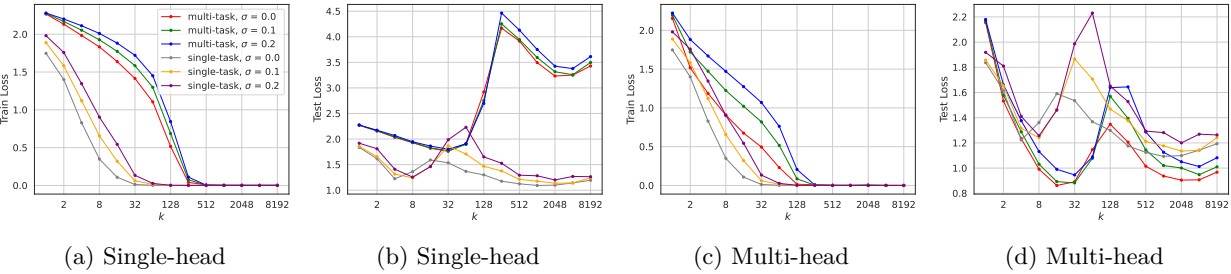

|(a) Single-head|(b) Single-head|(c) Multi-head|(d) Multi-head|

Figure 2: Comparing the performance of the single-task vs multi-task learners on CIFAR-100 using pretrained ResNet-18 backbone. The vertical axis is the cross-entropy loss and the horizontal axis is the width of the MLP on top of the backbone. $\sigma$ represents the noisy portion of the training set that was corrupted by randomly switching its labels. (a) and (b) correspond to the train and test loss with the single-head classifier, while (c) and (d) show the train and test loss of the multi-head architecture. Similar to linear models studied previously, deep MTL models also undergo the double descent behaviors with the test error peak depending on the characteristics of the task similarity and strength of the noise.

model's complexity by considering different values for the number of convolutional filters in the backbone. In particular, the ResNet-18 has layers with $[k, 2k, 4k, 8k]$ number of filters and we control the width of the backbone by changing the value of $k$. The typical ResNet-18 width has $k = 64$ filters. On top of this backbone, we use an MLP layer with a single hidden layer of size 128, followed by a classification head.

In all of the single-task experiments, we use a linear classification head as the final layer. However, we consider two architectures for the MTL case. In the multi-head architecture, we employ a separate linear classifier for each task while in the single-head architecture, we use a shared linear head for all tasks.

**Multi-task vs single-task learning.** Figure 2 presents the train and test behavior of a multi-task DNN compared to single-task models, when trained on CIFAR-100 using the pretrained ResNet18. As depicted in Figures 2a and 2b, train error is almost zero for heavily overparameterized models. However, the test error behavior is in line with our theoretical analysis and is similar to the linear models illustrated in Figure 1 from several aspects: (i) The training interpolation threshold and the corresponding test loss peak for STL and MTL models are observable at different locations, which is consistent with the observation 1. (ii) The error peak exists even with zero amount of label noise but is intensified with larger noise, which aligns with the observation 3. (iii) A descent floor exists in the overparameterized regime where the test loss is minimized, which is related to the observation 4. These observations are crucial when designing MTL models and should be considered when optimizing the model size.

In Figures 2a and 2b, while the single-head architecture is used and although the training error is zero for large enough models, we observe that single-task learners significantly outperform multi-task learners. Inspired by the observations 2 and 3, this behavior suggests that weight sharing is not happening effectively, and the tasks are not fully collaborative, which can be considered a type of negative transfer learning. To dig deeper into this issue, we analyze the optimal weight for each

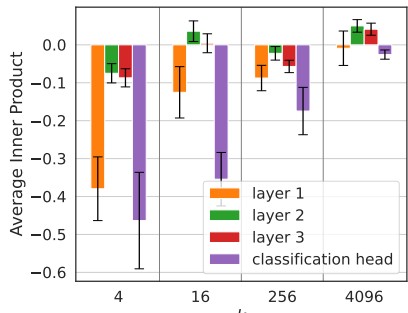

Figure 3: The average across-task inner product of task-specific optimal weights at different layers for different values of MLP width, $k$. As this figure shows, the weights of the classification head has the most negative inner products which means that the tasks are highly conflicting in this layer.

task per MLP layer. In other words, we start from a fully shared model, repeatedly pick and unshare the weights of a specific layer and optimize it to find the optimal weights of that layer per each task. Inspired by Equation 13, we calculate the inner product of the task-specific weights at each layer across different tasks and report the average values in Figure 3. For further details and discussion regarding this experiment, please refer to Appendix D.3.

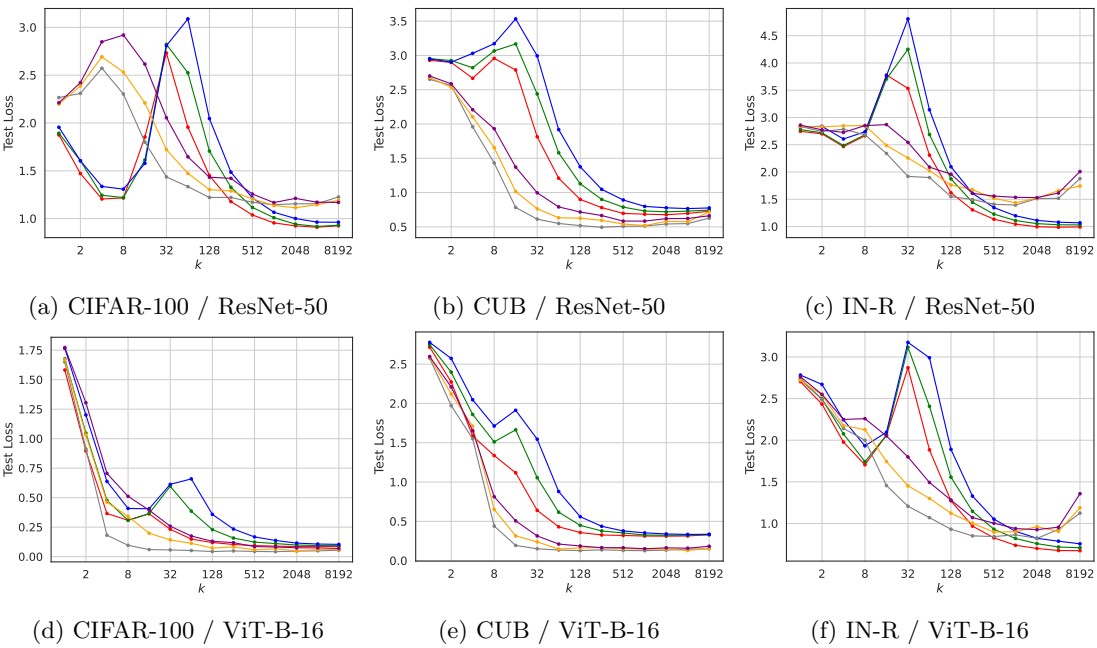

Figure 5: Test loss of the single-task vs multi-task learners on different datasets with pretrained backbones. The vertical axis is the cross-entropy loss and the horizontal axis is the width of the MLP on top of the backbone. $\sigma$ represents the noisy portion of the training set that was corrupted by randomly switching its labels. The multi-head architecture was used in all MTL experiments. These plots highlight the difference between MTL and STL double descent across several practical backbones and datasets.

According to Figure 3, we observe that the weights at the final classification layer have the most negative inner product values. This observation suggests that CIFAR-100 classes are highly conflicting in the classification layer. Nevertheless, extending the classification head to the multi-head architecture enables effective knowledge transfer in addition to an improved performance for the multi-task learner, as illustrated in Figure 2d. In other words, while unsharing any layer can benefit the MTL performance (as elaborated in Appendix D.3), unsharing the classification head is the most effective approach, which leads to a desired behavior closely similar to the collaborative version of the linear models in Figure 1a. As an additional experiment, we present Figure 4, which provides a similar comparison using an unfrozen version of ResNet-18 training from random initialization. Although this backbone is much deeper compared to the linear models we studied in Section 3.2, the consistent behavior suggests that our findings also generalize to more complex architectures.

Additionally, Figure 5 offers further experiments on CUB and IN-R datasets trained on two different pretrained backbones. Observing similar phenomena to what we presented in observations 1-3, endorses our hypothesis regarding the connection between our results for linear models and DNNs in an MTL setup. For example, in all of these experiments the test error peak of the multi-task learner is considerably strong, while in some configurations, the STL error peak is negligible. This observation emphasizes that the label noise is not the sole source of the error peak; rather, as predicted from our theoretical analysis, the inherent dissimilarity across different tasks will also contribute to the strength of the peak. We conclude that our results help explain

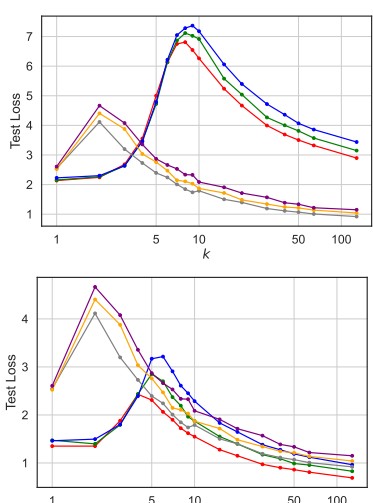

Figure 4: Test loss of the single-task (**Top**) vs multi-task (**Bottom**) learners on CIFAR-100 with unfrozen ResNet-18. The vertical axis is the cross-entropy loss and the horizontal axis is the scale factor that controls the number of filters in convolutional layers. As we observe here, the test loss behaves similarly to the case where only the MLP head is trained.

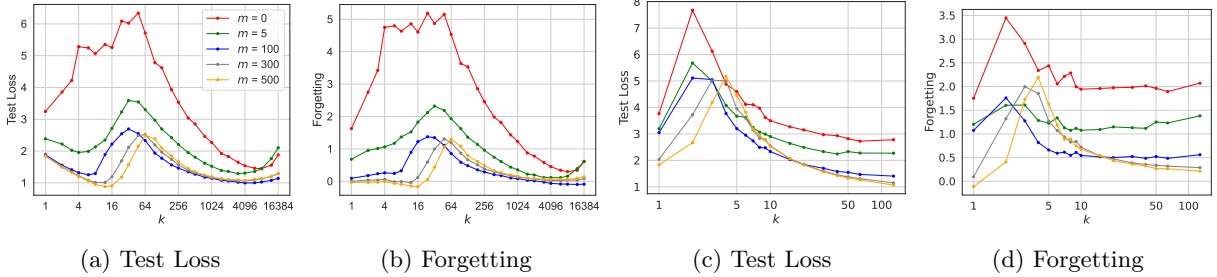

(a) Test Loss       (b) Forgetting       (c) Test Loss       (d) Forgetting

Figure 6: The test loss and forgetting of a continual learner on CIFAR-100 w.r.t. width of the model, $k$, for different levels of per-class memory capacity. $m = 0$ means naive fine-tuning, and $m = 500$ corresponds to the full MTL setup. (a) and (b) correspond to the experiments with the Pretrained ResNet-18 and subfigures (c) and (d) show the results when fine-tuning a ResNet-18 model from scratch. The multi-head architecture with no label noise was used in these experiments. These figures show that a CL learner also has a test error peak which is affected by the size of the memory buffer.

the behavior of DNNs, and combined with empirical investigation, they can be used for optimal design. Complementary experiments, as well as the full training and testing curves of MTL experiments, are provided in Appendix D.1.

**Replay-based continual learning.** Figure 6 presents the test loss and the forgetting of a DNN continual learner on CIFAR-100 for different amounts of memory budget and model size. These models follow a trend very similar to the linear models presented in Section 4. In particular, there exists a test error peak for all models, but its location and strength are controlled by the memory size. By increasing the memory size, the test error peak naturally shifts to the right with reduced intensity. Another observation aligning with the linear models is that the forgetting greatly reduces for large model sizes, even without any memory, a phenomenon also confirmed by recent studies on sequential fine-tuning CL (Goldfarb & Hand, 2023).

Additionally, it is noticeable that increasing the memory capacity does not always uniformly increase the performance for all values of model size. Especially, the memory size is less effective with larger models. This raises a fundamental question when designing CL models in practice. Given a fixed budget for physical memory, is it more efficient to spend it on increasing the model size or building a bigger sample memory? Table 1 contains the results of experiments with varying budgets dedicated to the replay memory buffer. As the results show, different optimal balance points exist for all datasets that maximize the average accuracy and minimize forgetting. This observation aligns with the recent CL methods, where an equal portion of the sample memory budget is devoted to specialized blocks (Zhou et al., 2023; Wang et al., 2022) or task-specific tokens (Douillard et al., 2022). Therefore, we propose that our theoretical findings in linear models can be applied to practical CL algorithms to gain a deeper understanding.

Table 1: The effect of varying sample memory budget and architecture memory budget over the average accuracy and average forgetting of overparameterized continual learners on two datasets. All values in this table are presented as percentages. The sample budget is linearly proportional to the number of training images stored in the memory, where 100% corresponds to storing 10% of the whole training set. A complete version of this table with more details is presented in Appendix C

| Mem. Budg. | Arch. Budg. | CIFAR-100 | | Imagenet-R | | CUB | | PMNIST | |
|---|---|---|---|---|---|---|---|---|---|
| | | Avg. Acc. | Avg. Forg. | Avg. Acc. | Avg. Forg. | Avg. Acc. | Avg. Forg. | Avg. Acc. | Avg. Forg. |
| 100 | <1 | $56.6_{\pm0.4}$ | $-22.5_{\pm0.5}$ | $76.1_{\pm0.4}$ | $-3.3_{\pm0.6}$ | $87.6_{\pm0.2}$ | $-4.5_{\pm0.2}$ | $32.8_{\pm2.8}$ | $-69.8_{\pm3.0}$ |
| 80 | 20 | $70.6_{\pm0.1}$ | $-7.1_{\pm0.1}$ | $76.9_{\pm0.6}$ | $-2.0_{\pm0.3}$ | $84.1_{\pm0.5}$ | $-6.7_{\pm0.0}$ | $82.9_{\pm1.0}$ | $-13.9_{\pm1.1}$ |
| 60 | 40 | $\mathbf{71.9_{\pm0.2}}$ | $\mathbf{-5.4_{\pm0.3}}$ | $77.1_{\pm0.6}$ | $-1.4_{\pm0.4}$ | $88.5_{\pm0.3}$ | $-1.8_{\pm0.6}$ | $87.7_{\pm0.4}$ | $-8.5_{\pm0.4}$ |
| 40 | 60 | $66.3_{\pm0.4}$ | $-11.3_{\pm0.4}$ | $\mathbf{78.9_{\pm0.5}}$ | $-1.0_{\pm0.1}$ | $\mathbf{92.0_{\pm0.1}}$ | $\mathbf{-0.1_{\pm0.2}}$ | $\mathbf{88.8_{\pm0.1}}$ | $\mathbf{-7.1_{\pm0.1}}$ |
| 20 | 80 | $67.4_{\pm0.3}$ | $-9.7_{\pm0.3}$ | $78.5_{\pm0.3}$ | $\mathbf{-0.6_{\pm0.1}}$ | $91.7_{\pm0.4}$ | $-0.4_{\pm0.3}$ | $88.2_{\pm0.3}$ | $-7.2_{\pm0.2}$ |
| 0 | 100 | $63.8_{\pm0.3}$ | $-10.8_{\pm0.3}$ | $49.5_{\pm0.6}$ | $-20.5_{\pm0.5}$ | $75.8_{\pm0.7}$ | $-12.2_{\pm0.7}$ | $68.2_{\pm1.0}$ | $-17.9_{\pm0.2}$ |

## 6 Conclusion

We studied overparameterized linear models in the MTL and CL settings and derived exact theoretical characterizations to better understand the impact of various system parameters over the average generalization error and knowledge transfer. We also analyzed replay-based continual learning and provided theoretical results to describe the generalization error and forgetting of such methods. We also performed extensive experiments with DNNs to show that similar behaviors are observable in the deep MTL and CL models, and understanding the linear overparameterized models can guide our intuition when studying more complex models. Please see Appendix E for a discussion on the limitations and future work.

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

# A   Proof of Theorems.

## A.1   Useful Lemmas

In this section, we start by providing some useful lemmas and then provide proofs for the theorems in the main text.

**Lemma A.1.** *Consider a square invertible matrix $S \in \mathbb{R}^{n \times n}$. Assume $S$ can be partitioned into four smaller blocks as*

$$S = \begin{bmatrix} A & B \\ C & D \end{bmatrix},$$

*where $A$ and $D$ are invertible square matrices with arbitrary relative sizes. Denote $H = D - CA^{-1}B$ and assume that $H$ is invertible. Then, the inverse of $S$ can be written as*

$$S^{-1} = \begin{bmatrix} A & B \\ C & D \end{bmatrix}^{-1} = \begin{bmatrix} A^{-1} + A^{-1}BH^{-1}CA^{-1} & -A^{-1}BH^{-1} \\ -H^{-1}CA^{-1} & H^{-1} \end{bmatrix},$$

*Proof.* The lemma can be proved by simply examining that $SS^{-1} = I$. □

**Lemma A.2.** *Consider a square invertible matrix $S \in \mathbb{R}^{n \times n}$. Assume $S$ can be partitioned into $3 \times 3$ blocks:*

$$S = \begin{bmatrix} E & F & G \\ H & J & K \\ L & M & N \end{bmatrix}.$$

*Then the inverse of $S$ is*

$$S^{-1} = \begin{bmatrix} E^{-1} + E^{-1}(FA^{-1}H + UR^{-1}V)E^{-1} & -E^{-1}(F - UR^{-1}C)A^{-1} & -E^{-1}UR^{-1} \\ -A^{-1}(H - BR^{-1}V)E^{-1} & A^{-1} + A^{-1}BR^{-1}CA^{-1} & -A^{-1}BR^{-1} \\ -R^{-1}VE^{-1} & -R^{-1}CA^{-1} & R^{-1} \end{bmatrix}$$

*where we define*

$$A = J - HE^{-1}F$$
$$B = K - HE^{-1}G$$
$$C = M - LE^{-1}F$$
$$D = N - LE^{-1}G$$
$$U = G - FA^{-1}B$$
$$V = L - CA^{-1}H$$
$$R = D - CA^{-1}B$$

*and we assume that the matrix inverses exist wherever necessary.*

*Proof.* Similarly this lemma is proved by examining $SS^{-1} = I$. □

**Notations.** We have already introduced some matrix notations in Section 3.1. In this section, we continue to introduce new helpful notations. Remember that for task $t \in [T]$ with $n_t$ samples, we had $X_t \in \mathbb{R}^{p \times n_t}$, $w_t^* \in \mathbb{R}^p$, $z_t \in \mathbb{R}^{n_t}$ and $y_t = X_t^\top w_t^* + z_t$.

Now consider indices $i, j \in [T]$ such that $i < j$ and denote $n_{ij} = \sum_{t=i}^{j} n_t$. We introduce the concatenated data matrix as $X_{ij} \in \mathbb{R}^{p \times n_{ij}}$ which is constructed by concatenating data matrices from $X_i$ to $X_j$ in the second dimension, i.e. $X_{ij} = \begin{bmatrix} X_i & X_{i+1} & ... & X_j \end{bmatrix}$. With this notation, the matrix representation of all training samples is $X_{1:T} \in \mathbb{R}^{p \times \bar{n}}$ where $\bar{n} = \sum_{t=1}^{T} n_t$. Additionally, we introduce the matrix $X_{0|t} \in \mathbb{R}^{p \times \bar{n}}$ which is built from $X_{1:T}$ by replacing all entries with zeros except at the columns corresponding to the task $t$. In other words, $X_{0|t} = \begin{bmatrix} 0_{p \times r} & X_t & 0_{p \times q} \end{bmatrix}$, where $0_{p \times r}$ is an all zero matrix with size $p \times r$, $r = \sum_{i=1}^{t-1} n_i$ and $q = \sum_{i=t+1}^{T} n_i$.

Since our study mainly focuses on overparameterized regimes, the data points $X_t \in \mathbb{R}^{p \times n_t}$ are random matrices with more rows than columns. Therefore, $(X_t^\top X_t)^{-1}$ exists almost surely and we define $X_t^\dagger = X_t(X_t^\top X_t)^{-1}$ and the projection matrix $P_t = X_t(X_t^\top X_t)^{-1} X_t^\top$. Additionally, let $P_{ij} = X_{ij}(X_{ij}^\top X_{ij})^{-1} X_{ij}^\top$, $X_{ij}^\dagger = X_{ij}(X_{ij}^\top X_{ij})^{-1}$, and $P_{0|t} = X_{1:T}(X_{1:T}^\top X_{1:T})^{-1} X_{0|t}^\top$. With this in mind, we provide some useful lemmas.[2]

**Lemma A.3.** *Let $X_{1:2} \in \mathbb{R}^{p \times (n_1+n_2)}$ be the result of concatenating $X_1 \in \mathbb{R}^{p \times n_1}$ and $X_2 \in \mathbb{R}^{p \times n_2}$. Also let $X_{0|1} \in \mathbb{R}^{p \times (n_1+n_2)}$ be result of concatenating $X_1$ with a zero matrix. Similarly, build $X_{0|2} \in \mathbb{R}^{p \times (n_1+n_2)}$ by concatenating a zero matrix with $X_2$. Assuming $p > n_1 + n_2$ and that all inverses exist, it holds that*

   *i.* $P_{1:2} = X_{1:2}(X_{1:2}^\top X_{1:2})^{-1} X_{1:2}^\top = P_1 + (I - P_1)X_2(X_2^\top(I - P_1)X_2)^{-1} X_2^\top(I - P_1)$

   *ii.* $P_{0|1} = X_{1:2}(X_{1:2}^\top X_{1:2})^{-1} X_{0|1}^\top = P_1 - (I - P_1)X_2(X_2^\top(I - P_1)X_2)^{-1} X_2^\top P_1$

   *iii.* $P_{0|2} = X_{1:2}(X_{1:2}^\top X_{1:2})^{-1} X_{0|2}^\top = (I - P_1)X_2(X_2^\top(I - P_1)X_2)^{-1} X_2^\top$

   *iv.* $P_{0|1}^\top P_{0|1} = X_{0|1}(X_{1:2}^\top X_{1:2})^{-1} X_{0|1}^\top = P_1 + P_1 X_2(X_2^\top(I - P_1)X_2)^{-1} X_2^\top P_1$

   *v.*
$$X_{1:2}^\dagger = X_{1:2}(X_{1:2}^\top X_{1:2})^{-1}$$
$$= \begin{bmatrix} X_1^\dagger - (I - P_1)X_2(X_2^\top(I - P_1)X_2)^{-1} X_2^\top X_1^\dagger & (I - P_1)X_2(X_2^\top(I - P_1)X_2)^{-1} \end{bmatrix}$$

*where $P_1 = X_1(X_1^\top X_1)^{-1} X_1^\top$, $X_1^\dagger = X_1(X_1^\top X_1)^{-1}$ and $I$ is the identity matrix with size $p \times p$.*

*Proof.* We start by rewriting $X_{1:2}^\top X_{1:2}$ as a block matrix and then finding its inverse:

$$(X_{1:2}^\top X_{1:2})^{-1} = \left( \begin{bmatrix} X_1^\top X_1 & X_1^\top X_2 \\ X_2^\top X_1 & X_2^\top X_2 \end{bmatrix} \right)^{-1}$$

Now using Lemma A.1

$$(X_{1:2}^\top X_{1:2})^{-1} = \begin{bmatrix} (X_1^\top X_1)^{-1} + (X_1^\top X_1)^{-1} X_1^\top X_2 H^{-1} X_2^\top X_1(X_1^\top X_1)^{-1} & -(X_1^\top X_1)^{-1} X_1^\top X_2 H^{-1} \\ -H^{-1} X_2^\top X_1(X_1^\top X_1)^{-1} & H^{-1} \end{bmatrix},$$

where $H = X_2^\top X_2 - X_2^\top X_1(X_1^\top X_1)^{-1} X_1^\top X_2 = X_2^\top(I - P_1)X_2$. Now, by noticing that $X_{1:2} = \begin{bmatrix} X_1 & X_2 \end{bmatrix}$, we can prove what was desired by doing multiplications:

---

[2]Notice that $P_{0|t}$, by definition is not a projection matrix and we use this notation just for consistency.

i.

$$P_{1:2} = \begin{bmatrix} X_1 & X_2 \end{bmatrix} \left( \begin{bmatrix} X_1^\top X_1 & X_1^\top X_2 \\ X_2^\top X_1 & X_2^\top X_2 \end{bmatrix} \right)^{-1} \begin{bmatrix} X_1^\top \\ X_2^\top \end{bmatrix}$$

$$= P_1 + P_1 X_2 H^{-1} X_2^\top P_1 - X_2 H^{-1} X_2^\top P_1 - P_1 X_2 H^{-1} X_2^\top + X_2 H^{-1} X_2^\top$$

$$= P_1 + (I - P_1) X_2 (X_2^\top (I - P_1) X_2)^{-1} X_2^\top (I - P_1)$$

ii.

$$P_{0|1} = \begin{bmatrix} X_1 & X_2 \end{bmatrix} \left( \begin{bmatrix} X_1^\top X_1 & X_1^\top X_2 \\ X_2^\top X_1 & X_2^\top X_2 \end{bmatrix} \right)^{-1} \begin{bmatrix} X_1^\top \\ 0 \end{bmatrix}$$

$$= P_1 + P_1 X_2 H^{-1} X_2^\top P_1 - X_2 H^{-1} X_2^\top P_1$$

$$= P_1 - (I - P_1) X_2 (X_2^\top (I - P_1) X_2)^{-1} X_2^\top P_1$$

iii.

$$P_{0|2} = \begin{bmatrix} X_1 & X_2 \end{bmatrix} \left( \begin{bmatrix} X_1^\top X_1 & X_1^\top X_2 \\ X_2^\top X_1 & X_2^\top X_2 \end{bmatrix} \right)^{-1} \begin{bmatrix} 0 \\ X_2^\top \end{bmatrix}$$

$$= -P_1 X_2 H^{-1} X_2^\top + X_2 H^{-1} X_2^\top$$

$$= (I - P_1) X_2 (X_2^\top (I - P_1) X_2)^{-1} X_2^\top$$

iv.

$$P_{0|1}^\top P_{0|1} = \begin{bmatrix} X_1 & 0 \end{bmatrix} \left( \begin{bmatrix} X_1^\top X_1 & X_1^\top X_2 \\ X_2^\top X_1 & X_2^\top X_2 \end{bmatrix} \right)^{-1} \begin{bmatrix} X_1^\top \\ 0 \end{bmatrix}$$

$$= P_1 + P_1 X_2 H^{-1} X_2^\top P_1 = P_1 + P_1 X_2 (X_2^\top (I - P_1) X_2)^{-1} X_2^\top P_1$$

v.

$$X_{1:2}^\dagger = \begin{bmatrix} X_1 & X_2 \end{bmatrix} \begin{bmatrix} (X_1^\top X_1)^{-1} + (X_1^\top X_1)^{-1} X_1^\top X_2 H^{-1} X_2^\top X_1^\dagger & -(X_1^\top X_1)^{-1} X_1^\top X_2 H^{-1} \\ -H^{-1} X_2^\top X_1 (X_1^\top X_1)^{-1} & H^{-1} \end{bmatrix}$$

$$= \begin{bmatrix} X_1^\dagger - (I - P_1) X_2 (X_2^\top (I - P_1) X_2)^{-1} X_2^\top X_1^\dagger & (I - P_1) X_2 (X_2^\top (I - P_1) X_2)^{-1} \end{bmatrix}$$

$\square$

**Lemma A.4.** *Let* $X_{1:3} \in \mathbb{R}^{p \times (n_1 + n_2 + n_3)}$ *be the result of concatenating* $X_1 \in \mathbb{R}^{p \times n_1}$, $X_2 \in \mathbb{R}^{p \times n_2}$ *and* $X_3 \in \mathbb{R}^{p \times n_3}$. *Also let* $X_{0|1}, X_{0|2}, X_{0|3} \in \mathbb{R}^{p \times (n_1 + n_2 + n_3)}$ *be the result of concatenating* $X_1$, $X_2$ *and* $X_3$ *with zero matrices such that* $X_{0|1} = \begin{bmatrix} X_1 & 0 & 0 \end{bmatrix}$, $X_{0|2} = \begin{bmatrix} 0 & X_2 & 0 \end{bmatrix}$ *and* $X_{0|3} = \begin{bmatrix} 0 & 0 & X_3 \end{bmatrix}$. *Define* $P_{0|1} = X_{1:3} (X_{1:3}^\top X_{1:3})^{-1} X_{0|1}^\top$ *and* $P_{0|3} = X_{1:3} (X_{1:3}^\top X_{1:3})^{-1} X_{0|3}^\top$. *Assuming* $p > n_1 + n_2 + n_3$ *and that all inverses exist, it holds that*

$$P_{0|1}^\top P_{0|3} = -P_1 (I - \hat{P}_{1:2}) X_3 (X_3^\top (I - P_1)(I - \hat{P}_{1:2}) X_3)^{-1} X_3^\top$$

*where* $P_1 = X_1 (X_1^\top X_1)^{-1} X_1^\top$ *and we denote* $\hat{P}_{1:2} = X_2 (X_2^\top (I - P_1) X_2)^{-1} X_2^\top (I - P_1)$.

*Proof.*

$$P_{0|1}^\top P_{0|3} = X_{0|1}(X_{1:3}^\top X_{1:3})^{-1}X_{1:3}^\top X_{1:3}(X_{1:3}^\top X_{1:3})^{-1}X_{0|3}^\top$$

$$= X_{0|1}(X_{1:3}^\top X_{1:3})^{-1}X_{0|3}^\top$$

$$= X_{0|1}\left(\begin{bmatrix} X_1^\top X_1 & X_1^\top X_2 & X_1^\top X_3 \\ X_2^\top X_1 & X_2^\top X_2 & X_2^\top X_3 \\ X_3^\top X_1 & X_3^\top X_2 & X_3^\top X_3 \end{bmatrix}\right)^{-1}X_{0|3}^\top$$

Now we use Lemma A.2 to find the matrix inverse. With the notation introduced in that lemma, we are looking for $-X_1 E^{-1}UR^{-1}X_3^\top$. Thus we have,

$$A = J - HE^{-1}F = X_2^\top(I - P_1)X_2$$

$$B = K - HE^{-1}G = X_2^\top(I - P_1)X_3$$

$$C = M - LE^{-1}F = X_3^\top(I - P_1)X_2$$

$$D = N - LE^{-1}G = X_3^\top(I - P_1)X_3$$

$$U = G - FA^{-1}B = X_1^\top(I - \hat{P}_{1:2})X_3$$

$$R = D - CA^{-1}B = X_3^\top(I - P_1)(I - \hat{P}_{1:2})X_3$$

$$\Rightarrow P_{0|1}^\top P_{0|3} = -X_1 E^{-1}UR^{-1}X_3^\top = -P_1(I - \hat{P}_{1:2})X_3(X_3^\top(I - P_1)(I - \hat{P}_{1:2})X_3)^{-1}X_3^\top$$

$\square$

**Lemma A.5.** *Assume $X \in \mathbb{R}^{p \times n}$ is the training data matrix and $y \in \mathbb{R}^n$ are the corresponding labels. In the underparameterized regime where $p < n$, we seek to solve an optimization of the form*

$$w^* = \underset{w \in \mathbb{R}^p}{\arg\min} \|X^\top w - y\|_2^2.$$

*The optimal solution to this optimization is given by*

$$w^* = (XX^\top)^{-1}Xy$$

*Proof.* By setting the derivative of the objective to zero, we obtain:

$$X(X^\top w^* - y) = 0 \Rightarrow w^* = (XX^\top)^{-1}Xy$$

noticing that $(XX^\top)^{-1}$ exist almost surely if $X$ is a Gaussian random matrix with $p < n$. $\square$

**Lemma A.6.** *Let $X \in \mathbb{R}^{p \times n}$ be the training data matrix and $y \in \mathbb{R}^n$ be the corresponding labels. Assume $w_0 \in \mathbb{R}^p$ is an arbitrary fixed vector. In the overparameterized regime where $p > n$, we seek to solve optimizations of the form*

$$w^* = \arg\min_{w \in \mathbb{R}^p} \|w - w_0\|_2^2 \quad s.t. \quad X^\top w = y.$$

*The optimal solution to this optimization is given by*

$$w^* = (I - P)w_0 + X^\dagger y$$

*where $P = X(X^\top X)^{-1} X^\top$ and $X^\dagger = X(X^\top X)^{-1}$.*

*Proof.* Using the Lagrange multipliers and by setting the derivatives to 0, we can get:

$$w^*, \lambda^* = \arg\min_{w,\lambda} \frac{1}{2}\|w - w_0\|_2^2 + \lambda^\top (X^\top w - y).$$

$$\Rightarrow w^* - w_0 + X\lambda^* = 0 \Rightarrow w^* = -X\lambda^* + w_0$$

$$X^\top w^* = y \Rightarrow -X^\top X \lambda^* + X^\top w_0 = y \Rightarrow \lambda^* = (X^\top X)^{-1} X^\top w_0 - (X^\top X)^{-1} y$$

$$\Rightarrow w^* = -X(X^\top X)^{-1} X^\top w_0 + X(X^\top X)^{-1} y + w_0 = (I - P)w_0 + X^\dagger y$$

$\square$

**Lemma A.7.** *Assume matrices $X_1 \in \mathbb{R}^{p \times n_1}$ and $X_2 \in \mathbb{R}^{p \times n_2}$ to be random matrices with entries being independently sampled from the standard Gaussian distribution $\mathcal{N}(0, 1)$. Assume $p > n_1 + n_2 + 1$ and let $P_1 = X_1(X_1^\top X_1)^{-1} X_1^\top$ be the orthogonal projection matrix that projects onto the column space of $X_1$. Also assume $X_{1:2} \in \mathbb{R}^{p \times (n_1+n_2)}$ be the result of concatenation of $X_1$ and $X_2$, and $X_{0|1} \in \mathbb{R}^{p \times (n_1+n_2)}$ be the result of concatenation of $X_1$ with a zero matrix. Additionally, let $P_{1:2} = X_{1:2}(X_{1:2}^\top X_{1:2})^{-1} X_{1:2}^\top$ and $P_{0|1} = X_{1:2}(X_{1:2}^\top X_{1:2})^{-1} X_{0|1}^\top$. Assuming $w \in \mathbb{R}^p$ is a fixed given vector, the following equalities hold:*

    *i.* $\mathbb{E}[\|P_1 w\|^2] = \frac{n_1}{p}\|w\|^2$

    *ii.* $\mathbb{E}[\|(I - P_1)w\|^2] = (1 - \frac{n_1}{p})\|w\|^2$

    *iii.* $\mathbb{E}[\|P_{0|1} w\|^2] = \frac{n_1}{p}(1 + \frac{n_2}{p - (n_1+n_2) - 1})\|w\|^2$

*Proof.*

  i. Without loss of generality, we can focus on finding $\mathbb{E}[\|P_1 u\|^2]$ where $u \in \mathbb{R}^p$ and $\|u\|^2 = 1$, since $X_1$ is a Gaussian matrix with rank $n_1$, and $P_1 = X_1(X_1^\top X_1)^{-1} X_1^\top$ is an orthogonal projection matrix with a similar rank. Due to the rotational invariance of the standard normal distribution, we can assume that $P_1$ is a fixed matrix and instead, $u$ is a random vector uniformly sampled from the unit sphere in $\mathbb{R}^p$. Using the rotational invariance again, we may assume without the loss of generality that $P_1$ is the coordinate projection onto the first $n_1$ coordinates in $\mathbb{R}^p$. Then it holds that:

$$\mathbb{E}_{X_1}[\|P_1 u\|^2] = \mathbb{E}_u[\sum_{i=1}^{n_1} u_i^2] = \frac{n_1}{p}$$

  ii. $I - P_1$ is a projection orthogonal to $P_1$. Therefore, by Pythagorean theorem,

$$\mathbb{E}[\|(I - P_1)w\|^2] = \|w\|^2 - \mathbb{E}[\|P_1 w\|^2] = (1 - \frac{n_1}{p})\|w\|^2$$

iii.

$$\mathbb{E}[\|P_{0|1} w\|^2] = \mathbb{E}[w^\top P_{0|1}^\top P_{0|1} w]$$

$$= \mathbb{E}[w^\top P_1 w] + \mathbb{E}[w^\top P_1 X_2 (X_2^\top (I - P_1) X_2)^{-1} X_2^\top P_1 w] \qquad \text{(Lemma A.3 part (iv))}$$

$$= \frac{n_1}{p}\|w\|^2 + \mathbb{E}[\text{tr}(w^\top P_1 X_2 (X_2^\top (I - P_1) X_2)^{-1} X_2^\top P_1 w)]$$

$$= \frac{n_1}{p}\|w\|^2 + \mathbb{E}_{X_1}[\text{tr}(\mathbb{E}_{X_2}[X_2^\top P_1 w w^\top P_1 X_2 (X_2^\top (I - P_1) X_2)^{-1}])]$$

To find the above expectation over $X_2$, notice that $P_1$ is an orthogonal projection matrix and, therefore, for a fixed $P_1$, $P_1 X_2$ is independent of $(I - P_1)X_2$.

$$\Rightarrow \mathbb{E}[\|P_{0|1} w\|^2] = \frac{n_1}{p}\|w\|^2 + \mathbb{E}_{X_1}[\text{tr}(\mathbb{E}_{X_2}[X_2^\top P_1 w w^\top P_1 X_2] \mathbb{E}_{X_2}[(X_2^\top (I - P_1) X_2)^{-1}])]$$

$$= \frac{n_1}{p}\|w\|^2 + \mathbb{E}_{X_1}[\text{tr}(\text{tr}(P_1 w w^\top P_1) \mathbb{E}_{X_2}[(X_2^\top (I - P_1) X_2)^{-1}])]$$

$$= \frac{n_1}{p}\|w\|^2 + \mathbb{E}_{X_1}[\|P_1 w\|^2 \, \text{tr}(\mathbb{E}_{X_2}[(X_2^\top (I - P_1) X_2)^{-1}])]$$

We focus on finding the inner expectation first. Notice that for a fixed $X_1$, $I - P_1$ is an orthogonal projection matrix with rank $p-n_1$. Due to the rotational invariance of the standard normal distribution, we may assume without loss of generality that $I - P_1$ is the projection matrix that projects onto the first $p - n_1$ coordinates. With this in mind, $(X_2^\top (I - P_1) X_2)^{-1}$ follows an inverse-Wishart distribution with an identity scale matrix $I_{n_2 \times n_2} \in \mathbb{R}^{n_2}$ and $p - n_1$ degrees of freedom.

$$\Rightarrow \mathbb{E}[\|P_{0|1} w\|^2] = \frac{n_1}{p} \|w\|^2 + \mathbb{E}_{X_1}[\|P_1 w\|^2 \operatorname{tr}(\frac{I_{n_2 \times n_2}}{p - n_1 - n_2 - 1})]$$

$$= \frac{n_1}{p} \|w\|^2 + \frac{n_2}{p - (n_1 + n_2) - 1} \mathbb{E}_{X_1}[\|P_1 w\|^2]$$

$$= \frac{n_1}{p}(1 + \frac{n_2}{p - (n_1 + n_2) - 1}) \|w\|^2$$

$\square$

**Lemma A.8.** *Assume matrices $X_1 \in \mathbb{R}^{p \times n_1}$, $X_2 \in \mathbb{R}^{p \times n_2}$ and $X_3 \in \mathbb{R}^{p \times n_3}$ to be random matrices with entries being independently sampled from the standard Gaussian distribution $\mathcal{N}(0, 1)$. Assume $p > n_1 + n_2 + n_3 + 1$ and let $X_{1:3} \in \mathbb{R}^{p \times (n_1 + n_2 + n_3)}$ be the result of concatenation of $X_1$, $X_2$, $X_3$. Also let $X_{0|1}, X_{0|2}, X_{0|3} \in \mathbb{R}^{p \times (n_1 + n_2 + n_3)}$ be the result of concatenating $X_1$, $X_2$ and $X_3$ with zero matrices such that $X_{0|1} = \begin{bmatrix} X_1 & 0 & 0 \end{bmatrix}$, $X_{0|2} = \begin{bmatrix} 0 & X_2 & 0 \end{bmatrix}$ and $X_{0|3} = \begin{bmatrix} 0 & 0 & X_3 \end{bmatrix}$. Define $P_{0|1} = X_{1:3}(X_{1:3}^\top X_{1:3})^{-1} X_{0|1}^\top$ and $P_{0|3} = X_{1:3}(X_{1:3}^\top X_{1:3})^{-1} X_{0|3}^\top$. Assuming $w, w' \in \mathbb{R}^p$ are fixed given vectors, it holds that*

$$\mathbb{E}[w^\top P_{0|1}^\top P_{0|3} w'] = -\frac{n_1 n_3}{p(p - (n_1 + n_2 + n_3) - 1)} \langle w, w' \rangle$$

*Proof.* Based on Lemma A.4, we have

$$\mathbb{E}[w^\top P_{0|1}^\top P_{0|3} w'] = -\mathbb{E}[w^\top P_1 (I - \hat{P}_{1:2}) X_3 (X_3^\top (I - P_1)(I - \hat{P}_{1:2}) X_3)^{-1} X_3^\top w']$$

$$= -\mathbb{E}_{X_1, X_2}[\mathbb{E}_{X_3}[w^\top P_1 (I - \hat{P}_{1:2}) X_3 (X_3^\top (I - P_1)(I - \hat{P}_{1:2}) X_3)^{-1} X_3^\top w']]$$

where $\hat{P}_{1:2} = X_2(X_2^\top (I - P_1) X_2)^{-1} X_2^\top (I - P_1)$. Now denote $P = (I - P_1)(I - \hat{P}_{1:2})$. With a fixed $X_1$ and $X_2$, $P$ is an orthogonal projection matrix, meaning that $P^\top = P^2 = P$. Therefore,

$$\mathbb{E}[w^\top P_{0|1}^\top P_{0|3} w'] = -\mathbb{E}_{X_1, X_2}[\mathbb{E}_{X_3}[w^\top P_1 (I - \hat{P}_{1:2}) X_3 (X_3^\top P X_3)^{-1} X_3^\top (P + I - P) w']]$$

$$= -\mathbb{E}_{X_1, X_2}[\mathbb{E}_{X_3}[w^\top P_1 (I - \hat{P}_{1:2}) X_3 (X_3^\top P X_3)^{-1} X_3^\top P w']]$$

$$- \mathbb{E}_{X_1, X_2}[\mathbb{E}_{X_3}[w^\top P_1 (I - \hat{P}_{1:2}) X_3 (X_3^\top P X_3)^{-1} X_3^\top (I - P) w']]$$

Notice that for a fixed $X_1$ and $X_2$, $P_1$ and $P$ are two orthogonal projection matrices. Therefore, the following independence relations hold:

$$(I - P) X_3 \perp\!\!\!\perp P X_3$$

$$P_1 (I - \hat{P}_{1:2}) X_3 \perp\!\!\!\perp (I - P_1)(I - \hat{P}_{1:2}) X_3$$

With this in mind, we can write

$$\mathbb{E}[w^\top P_{0|1}^\top P_{0|3} w'] = -\mathbb{E}_{X_1, X_2}[w^\top P_1 \mathbb{E}_{X_3}[(I - \hat{P}_{1:2}) X_3] \mathbb{E}_{X_3}[(X_3^\top P X_3)^{-1} X_3^\top P w']]$$

$$- \mathbb{E}_{X_1, X_2}[\mathbb{E}_{X_3}[\text{tr}(X_3^\top (I - P) w' w^\top P_1 (I - \hat{P}_{1:2}) X_3 (X_3^\top P X_3)^{-1})])]$$

$$= 0 - \mathbb{E}_{X_1, X_2}[\text{tr}(\mathbb{E}_{X_3}[X_3^\top (I - P) w' w^\top P_1 (I - \hat{P}_{1:2}) X_3] \mathbb{E}_{X_3}[(X_3^\top P X_3)^{-1}])]$$

$$= -\mathbb{E}_{X_1, X_2}[\text{tr}((I - P) w' w^\top P_1 (I - \hat{P}_{1:2})) \text{tr}(\mathbb{E}_{X_3}[(X_3^\top P X_3)^{-1}])]$$

Here we again use a technique similar to the one we used in Lemma A.7 part (iii) by first focusing on the inner expectation. Assume a fixed $X_1$ and $X_2$. Then $I - P_1$ and $P_1$ are orthogonal projections with ranks $p - n_1$ and $n_1$ respectively. Additionally, $(I - P_1)\hat{P}_{1:2}$ is an orthogonal projection matrix with rank $n_2$ and $p - n_1 > n_2$. On the other hand, $P_1$ and $(I - P_1)\hat{P}_{1:2}$ are orthogonal projections. Therefore, $P_1 + (I - P_1)\hat{P}_{1:2}$ is a projection matrix with rank $n_1 + n_2$ and $P = I - (P_1 + (I - P_1)\hat{P}_{1:2})$ is a projection matrix with rank $p - (n_1 + n_2)$. When taking the inner expectation only w.r.t. $X_3$, due to the rotational invariance property, we may assume that $(I - P_1)(I - \hat{P}_{1:2})$ is the coordinate projection onto the first $p - (n_1 + n_2)$ coordinates. Therefore, $(X_3^\top (I - P_1)(I - \hat{P}_{1:2}) X_3)^{-1} \sim \mathcal{IW}(I_{n_3 \times n_3}, p - (n_1 + n_2))$ and we have:

$$\Rightarrow \mathbb{E}[w^\top P_{0|1}^\top P_{0|3} w'] = -\mathbb{E}_{X_1, X_2}[\text{tr}((I - P) w' w^\top P_1 (I - \hat{P}_{1:2})) \text{tr}(\frac{I_{n_3 \times n_3}}{p - (n_1 + n_2) - n_3 - 1})]$$

$$= -\frac{n_3}{p - (n_1 + n_2 + n_3) - 1} \mathbb{E}_{X_1, X_2}[\text{tr}(w^\top P_1 (I - \hat{P}_{1:2})(I - P) w')]$$

Now, using the definition of $P$ and $\hat{P}_{1:2}$, we have

$$\mathbb{E}[w^\top P_{0|1}^\top P_{0|3} w'] = -\frac{n_3}{p - (n_1 + n_2 + n_3) - 1} \mathbb{E}[\text{tr}(w^\top P_1 (I - \hat{P}_{1:2}) w')]$$

$$= -\frac{n_3}{p - (n_1 + n_2 + n_3) - 1} \mathbb{E}[\text{tr}(w^\top P_1 w')]$$

$$+ \frac{n_3}{p - (n_1 + n_2 + n_3) - 1} \mathbb{E}[\text{tr}(w^\top P_1 X_2 (X_2^\top (I - P_1) X_2)^{-1} X_2^\top (I - P_1) w')]$$

Remember that for a given $X_1$, $P_1 X_2$ and $(I - P_1) X_2$ are independent. Therefore,

$$\mathbb{E}[w^\top P_{0|1}^\top P_{0|3} w'] = -\frac{n_3}{p - (n_1 + n_2 + n_3) - 1} \mathbb{E}[\text{tr}(w^\top P_1 w')]$$

$$= -\frac{n_3}{p - (n_1 + n_2 + n_3) - 1} \times \frac{1}{4} \mathbb{E}[(w + w')^\top P_1 (w + w') - (w - w')^\top P_1 (w - w')]$$

$$= -\frac{n_3}{p - (n_1 + n_2 + n_3) - 1} \times \frac{1}{4} \mathbb{E}[\|P_1 (w + w')\|^2 - \|P_1 (w - w')\|^2]$$

$$= -\frac{n_1 n_3}{p - (n_1 + n_2 + n_3) - 1} \times \frac{1}{4} (\|w + w'\|^2 - \|w - w'\|^2) \qquad \text{(Lemma A.7 part (i))}$$

$$= -\frac{n_3 n_1}{p(p - (n_1 + n_2 + n_3) - 1)} \langle w, w' \rangle$$

$\square$

As a concluding remark to this section, notice that When $X$ is a Gaussian matrix with iid entries, $X^T X$ is almost surely invertible (i.e. the probability measure of $X^T X$ not being invertible is zero). Therefore, in what comes next, we assume that all Gaussian matrices in this form are invertible without loss of generality.

## A.2 Proof of Theorems

Now that we have all the required building blocks, we are ready to provide the proofs. We start by proving the theorem related to the single-task learner and continue by proving the theorems on multi-task learning and continual learning.

### A.2.1 Proof of Theorem 3.1

*Proof.* **underparameterized regime.** In the underparameterized regime where $n_t \geq p + 2$, the signle-task learner is the unique solution to the Equation 4 and is obtained by $w_t = (X_t X_t^\top)^{-1} X_t y_t$ according to Lemma A.5. Therefore, we must have

$$w_t = (X_t X_t^\top)^{-1} X_t y_t = (X_t X_t^\top)^{-1} X_t (X_t^\top w_t^* + z_t) = w_t^* + (X_t X_t^\top)^{-1} X_t z_t$$

$$\Rightarrow \mathbb{E}[\mathcal{L}_t(w)] = \mathbb{E}[\|w_t - w_t^*\|^2] = \mathbb{E}[\|(X_t X_t^\top)^{-1} X_t z_t\|^2]$$

where the expectation is due to randomness of both $X_t$ and $z_t$. First, we take the expectation w.r.t. $z_t \sim \mathcal{N}_{n_t}(0, \sigma^2 I_{n_t \times n_t})$:

$$\begin{aligned}
\mathbb{E}[\mathcal{L}_t(w)] &= \mathbb{E}[z_t^\top X_t^\top (X_t X_t^\top)^{-1} (X_t X_t^\top)^{-1} X_t z_t] \\
&= \sigma^2 \mathbb{E}[\operatorname{tr}(X_t^\top (X_t X_t^\top)^{-2} X_t)] \\
&= \sigma^2 \mathbb{E}[\operatorname{tr}((X_t X_t^\top)^{-2} X_t X_t^\top)] \\
&= \sigma^2 \mathbb{E}[\operatorname{tr}((X_t X_t^\top)^{-1})] \\
&= \sigma^2 \operatorname{tr}(\mathbb{E}[(X_t X_t^\top)^{-1}])) \\
&= \sigma^2 \operatorname{tr}(\frac{I_{p \times p}}{n_t - p - 1}) \\
&= \frac{p \sigma^2}{n_t - p - 1}
\end{aligned}$$

where the two last line comes from the fact that $(X_t X_t^\top)^{-1}$ follows the inverse-Wishart distribution as $\mathcal{IW}(I_{p \times p}, n)$.

**overparameterized regime.** In the overparameterized regime where $p \geq n_t + 2$, we look for the solution of optimization 5. Therefore, based on Lemma A.6, we can have

$$
\begin{aligned}
\mathbb{E}[\mathcal{L}_t(w)] &= \mathbb{E}[\|w_t - w_t^*\|^2] \\
&= \mathbb{E}[\|X_t^\dagger y_t - w_t^*\|^2] \\
&= \mathbb{E}[\|X_t^\dagger (X_t^\top w_t^* + z_t) - w_t^*\|^2] \\
&= \mathbb{E}[\|(I - P_t)w_t^* - X_t^\dagger z_t\|^2] \\
&= \mathbb{E}[\|(I - P_t)w_t^*\|^2] + \mathbb{E}[\|X_t^\dagger z_t\|^2] - 2w_t^{*T}\mathbb{E}[(I - P_t)X_t^\dagger]\mathbb{E}[z_t] \\
&= \mathbb{E}[\|(I - P_t)w_t^*\|^2] + \mathbb{E}[z_t^\top (X_t^\top X_t)^{-1} z_t] && (\mathbb{E}[z_t] = 0) \\
&= \mathbb{E}[\|(I - P_t)w_t^*\|^2] + \frac{n_t \sigma^2}{p - n_t - 1} && ((X_t^\top X_t)^{-1} \sim \mathcal{IW}(I_{n_t \times n_t}, p)) \\
&= (1 - \frac{n_t}{p})\|w_t^*\|^2 + \frac{n_t \sigma^2}{p - n_t - 1} && (\text{Lemma A.7 part (ii)})
\end{aligned}
$$

$\square$

### A.2.2 Proof of Theorem 3.2

*Proof.* Based on Lemma A.6 we can write

$$w_{1:T} = X_{1:T}(X_{1:T}^\top X_{1:T})^{-1} y_{1:T}$$

Using the notations introduced in the previous section, we have

$$w_{1:T} = X_{1:T}(X_{1:T}^\top X_{1:T})^{-1}(\sum_{s=1}^{T} X_{0|s}^\top w_s^* + z_{1:T}) = \sum_{s=1}^{T} P_{0|s} w_s^* + X_{1:T}^\dagger z_{1:T}$$

Now we start by calculating the multi-task learner's loss for the task $i$:

$$\mathbb{E}[\mathcal{L}_i(w_{1:T})] = \mathbb{E}[\|w_{1:T} - w_i^*\|^2]$$

$$= \mathbb{E}[\|\sum_{s=1}^{T} P_{0|s} w_s^* + X_{1:T}^\dagger z_{1:T} - w_i^*\|^2]$$

$$= \mathbb{E}[\|\sum_{s=1}^{T} P_{0|s}(w_s^* - w_i^*) + (I - P_{1:T})w_i^* + X_{1:T}^\dagger z_{1:T}\|^2]$$

$$= \mathbb{E}[\|\sum_{s=1}^{T} P_{0|s}(w_s^* - w_i^*) + (I - P_{1:T})w_i^*\|^2] + \mathbb{E}[\|X_{1:T}^\dagger z_{1:T}\|^2] \qquad (z_{1:T} \perp\!\!\!\perp X_{1:T}, \mathbb{E}[z_{1:T}] = 0)$$

$$= \mathbb{E}[\|\sum_{s=1}^{T} P_{0|s}(w_s^* - w_i^*) + (I - P_{1:T})w_i^*\|^2] + \sigma^2 \operatorname{tr}(\mathbb{E}[(X_{1:T}X_{1:T})^{-1}]) \qquad (z_{1:T} \sim \mathcal{N}(0, \sigma^2 I))$$

$$= \mathbb{E}[\|\sum_{s=1}^{T} P_{0|s}(w_s^* - w_i^*) + (I - P_{1:T})w_i^*\|^2] + \frac{\bar{n}\sigma^2}{p - \bar{n} - 1} \qquad ((X_{1:T}X_{1:T})^{-1} \sim \mathcal{IW}(I_{\bar{n}\times\bar{n}}, p))$$

Therefore, we focus on finding the first term by expanding it. First notice that

$$(I - P_{1:T})P_{0|s} = (I - X_{1:T}(X_{1:T}^\top X_{1:T})^{-1} X_{1:T}^\top)X_{1:T}(X_{1:T}^\top X_{1:T})^{-1} X_{0|s}^\top$$

$$= X_{1:T}(X_{1:T}^\top X_{1:T})^{-1} X_{0|s}^\top - X_{1:T}(X_{1:T}^\top X_{1:T})^{-1} X_{0|s}^\top = 0$$

Thus,

$$\mathbb{E}[\|\sum_{s=1}^{T} P_{0|s}(w_s^* - w_i^*) + (I - P_{1:T})w_i^*\|^2]$$

$$= \mathbb{E}[\|\sum_{s=1}^{T} P_{0|s}(w_s^* - w_i^*)\|^2] + \mathbb{E}[\|(I - P_{1:T})w_i^*\|^2]$$

$$= \mathbb{E}[\|\sum_{s=1}^{T} P_{0|s}(w_s^* - w_i^*)\|^2] + (1 - \frac{\bar{n}}{p})\|w_i^*\|^2 \qquad \text{(Lemma A.7 part (ii))}$$

$$= \sum_{s=1}^{T} \mathbb{E}[\|P_{0|s}(w_s^* - w_i^*)\|^2] + \sum_{s=1}^{T} \sum_{\substack{s'=1 \\ s' \neq s}}^{T} \mathbb{E}[(w_s^* - w_i^*)^\top P_{0|s}^\top P_{0s'}(w_{s'}^* - w_i^*)] + (1 - \frac{\bar{n}}{p})\|w_i^*\|^2$$

Since the distribution of $P_{0|s}$ is invariant to the permutation of columns of $X_{1:T}$, we can focus on finding $\mathbb{E}[\|P_{0|1}(w_1^* - w_i^*)\|^2]$ and $\mathbb{E}[(w_1^* - w_i^*)^\top P_{0|1}^\top P_{0|T}(w_T^* - w_i^*)]$ without loss of generality. In Lemmas A.7 and A.8, we have already calculated similar quantities. Therefore,

$$\mathbb{E}[\|P_{0|1}(w_1^* - w_i^*)\|^2] = \frac{n_1}{p}(1 + \frac{\bar{n} - n_1}{p - \bar{n} - 1})\|w_1^* - w_i^*\|^2$$

and,

$$\mathbb{E}[(w_1^* - w_i^*)^\top P_{0|1}^\top P_{0|T}(w_T^* - w_i^*)] = -\frac{n_T n_1}{p(p - \bar{n} - 1)}\langle w_1^* - w_i^*, w_T^* - w_i^* \rangle$$

Overall, we can write :

$$\mathbb{E}[\mathcal{L}_i(w_{1:T})]$$

$$= -\frac{1}{p}\sum_{s=1}^{T}\sum_{\substack{s'=1\\s'\neq s}}^{T}\frac{n_s n_{s'}}{p - \bar{n} - 1}\langle w_s^* - w_i^*, w_{s'}^* - w_i^* \rangle + \sum_{s=1}^{T}\frac{n_s}{p}(1 + \frac{\bar{n} - n_s}{p - \bar{n} - 1})\|w_s^* - w_i^*\|^2$$

$$+ (1 - \frac{\bar{n}}{p})\|w_i^*\|^2 + \frac{\bar{n}\sigma^2}{p - \bar{n} - 1}$$

$$= -\frac{1}{p}\sum_{s=1}^{T}\sum_{s'=1}^{T}\frac{n_s n_{s'}}{p - \bar{n} - 1}\langle w_s^* - w_i^*, w_{s'}^* - w_i^* \rangle + \sum_{s=1}^{T}\frac{n_s}{p}(1 + \frac{\bar{n}}{p - \bar{n} - 1})\|w_s^* - w_i^*\|^2$$

$$+ (1 - \frac{\bar{n}}{p})\|w_i^*\|^2 + \frac{\bar{n}\sigma^2}{p - \bar{n} - 1}$$

$$= \frac{1}{2p}\sum_{s=1}^{T}\sum_{s'=1}^{T}\frac{n_s n_{s'}}{p - \bar{n} - 1}(\|w_s^* - w_{s'}^*\|^2 - \|w_s^* - w_i^*\|^2 - \|w_{s'}^* - w_i^*\|^2)$$

$$+ \sum_{s=1}^{T}\frac{n_s}{p}(1 + \frac{\bar{n}}{p - \bar{n} - 1})\|w_s^* - w_i^*\|^2 + (1 - \frac{\bar{n}}{p})\|w_i^*\|^2 + \frac{\bar{n}\sigma^2}{p - \bar{n} - 1}$$

$$= \frac{1}{2p}\sum_{s=1}^{T}\sum_{s'=1}^{T}\frac{n_s n_{s'}}{p - \bar{n} - 1}\|w_s^* - w_{s'}^*\|^2 + \sum_{s=1}^{T}\frac{n_s}{p}\|w_s^* - w_i^*\|^2 + (1 - \frac{\bar{n}}{p})\|w_i^*\|^2 + \frac{\bar{n}\sigma^2}{p - \bar{n} - 1} \qquad (21)$$

Now we calculate the desired metrics as

$$\mathbb{E}[G(w_{1:T})] = \frac{1}{T}\sum_{i=1}^{T}\mathbb{E}[\mathcal{L}_i(w_{1:T})]$$

$$= \frac{1}{2p}\sum_{s=1}^{T}\sum_{s'=1}^{T}\frac{n_s n_{s'}}{p - \bar{n} - 1}\|w_s^* - w_{s'}^*\|^2 + \frac{1}{T}\sum_{i=1}^{T}\sum_{s=1}^{T}\frac{n_s}{p}\|w_s^* - w_i^*\|^2$$

$$+ \frac{1}{T}\sum_{i=1}^{T}(1 - \frac{\bar{n}}{p})\|w_i^*\|^2 + \frac{\bar{n}\sigma^2}{p - \bar{n} - 1}$$

$$= \frac{1}{T}\sum_{i=1}^{T}\sum_{s=1}^{T}\frac{n_s}{p}(1 + \frac{\frac{T}{2}n_i}{p - \bar{n} - 1})\|w_s^* - w_i^*\|^2 + \frac{1}{T}(1 - \frac{\bar{n}}{p})\sum_{i=1}^{T}\|w_i^*\|^2 + \frac{\bar{n}\sigma^2}{p - \bar{n} - 1}$$

and

$$\mathbb{E}[K(w_{1:T})] = \frac{1}{T}\sum_{i=1}^{T}\mathbb{E}[\mathcal{L}_i(w_i)] - \mathbb{E}[\mathcal{L}_i(w_{1:T})]$$

$$= \frac{1}{T}\sum_{i=1}^{T}(1-\frac{n_i}{p})\|w_i^*\|^2 + \frac{1}{T}\sum_{i=1}^{T}\frac{n_i\sigma^2}{p-n_i-1}$$

$$- \frac{1}{T}\sum_{i=1}^{T}\sum_{s=1}^{T}\frac{n_s}{p}(1+\frac{\frac{T}{2}n_i}{p-\bar{n}-1})\|w_s^*-w_i^*\|^2 - \frac{1}{T}(1-\frac{\bar{n}}{p})\sum_{i=1}^{T}\|w_i^*\|^2 - \frac{\bar{n}\sigma^2}{p-\bar{n}-1}$$

$$= \frac{1}{T}\sum_{i=1}^{T}(1-\frac{n_i}{p})\|w_i^*\|^2 + \frac{1}{T}\sum_{i=1}^{T}\frac{n_i\sigma^2}{p-n_i-1}$$

$$- \frac{1}{T}\sum_{i=1}^{T}\sum_{s=1}^{T}\frac{n_s}{p}(1+\frac{\frac{T}{2}n_i}{p-\bar{n}-1})(\|w_s^*\|^2 + \|w_i^*\|^2 - 2\langle w_s^*, w_i^*\rangle)$$

$$- \frac{1}{T}(1-\frac{\bar{n}}{p})\sum_{i=1}^{T}\|w_i^*\|^2 - \frac{\bar{n}\sigma^2}{p-\bar{n}-1}$$

$$= \frac{2}{T}\sum_{i=1}^{T}\sum_{s=1}^{T}\frac{n_s}{p}(1+\frac{\frac{T}{2}n_i}{p-\bar{n}-1})\langle w_s^*, w_i^*\rangle - \frac{1}{T}\sum_{s=1}^{T}\frac{n_s}{p}(T+\frac{\frac{T}{2}\bar{n}}{p-\bar{n}-1})\|w_s^*\|^2$$

$$- \frac{1}{T}\sum_{i=1}^{T}\frac{\bar{n}}{p}(1+\frac{\frac{T}{2}n_i}{p-\bar{n}-1})\|w_i^*\|^2 - \frac{1}{T}(1-\frac{\bar{n}}{p})\sum_{i=1}^{T}\|w_i^*\|^2 - \frac{\bar{n}\sigma^2}{p-\bar{n}-1}$$

$$+ \frac{1}{T}\sum_{i=1}^{T}(1-\frac{n_i}{p})\|w_i^*\|^2 + \frac{1}{T}\sum_{i=1}^{T}\frac{n_i\sigma^2}{p-n_i-1}$$

$$= \frac{2}{T}\sum_{i=1}^{T}\sum_{s=1}^{T}\frac{n_s}{p}(1+\frac{\frac{T}{2}n_i}{p-\bar{n}-1})\langle w_s^*, w_i^*\rangle$$

$$- \frac{1}{T}\sum_{s=1}^{T}(\frac{Tn_s}{p}+\frac{n_s}{p}\frac{\frac{T}{2}\bar{n}}{p-\bar{n}-1}+\frac{\bar{n}}{p}\frac{\frac{T}{2}n_s}{p-\bar{n}-1}+\frac{n_s}{p})\|w_s^*\|^2$$

$$+ \frac{1}{T}\sum_{i=1}^{T}\frac{n_i\sigma^2}{p-n_i-1} - \frac{\bar{n}\sigma^2}{p-\bar{n}-1}$$

$$= \frac{2}{T}\sum_{i=1}^{T}\sum_{s=1}^{T}\frac{n_s}{p}(1+\frac{\frac{T}{2}n_i}{p-\bar{n}-1})\langle w_s^*, w_i^*\rangle - \sum_{s=1}^{T}\frac{n_s}{p}(1+\frac{1}{T}+\frac{\bar{n}}{p-\bar{n}-1})\|w_s^*\|^2$$

$$+ \frac{1}{T}\sum_{i=1}^{T}\frac{n_i\sigma^2}{p-n_i-1} - \frac{\bar{n}\sigma^2}{p-\bar{n}-1}$$

$\square$

### A.3  Proof of Theorem 4.1

*Proof.* To provide the proof, we start by finding the loss of task $i$ at timestep $t$, i.e. $\mathbb{E}[\mathcal{L}_i(w_{\vec{t}})]$, using an intermediate result in the proof provided in section A.2.2. In fact, we refer to Equation 21 and assume $n_1 = n_2 = ... = n_{t-1} = m$, $n_t = n$, $n_{t+1} = ... = n_T = 0$ and denote $\bar{n}_t = (t-1)m + n$ to achieve:

$$\mathbb{E}[\mathcal{L}_i(w_{\vec{t}})] = \frac{1}{2p}\sum_{s=1}^{t-1}\sum_{s'=1}^{t-1}\frac{m^2}{p - \bar{n}_t - 1}\|w_s^* - w_{s'}^*\|^2 + \frac{1}{p}\sum_{s=1}^{t-1}\frac{mn}{p - \bar{n}_t - 1}\|w_t^* - w_s^*\|^2$$

$$+ \sum_{s=1}^{t-1}\frac{m}{p}\|w_s^* - w_i^*\|^2 + \frac{n}{p}\|w_t^* - w_i^*\|^2 + (1 - \frac{\bar{n}_t}{p})\|w_i^*\|^2 + \frac{\bar{n}_t\sigma^2}{p - \bar{n}_t - 1} \tag{22}$$

.

Next, we set $T = 2$ in Equation 22 to get:

$$\mathbb{E}[\mathcal{L}_1(w_{\vec{2}})] = \frac{n}{p}(1 + \frac{m}{(p - (m+n) - 1)})\|w_1^* - w_2^*\|^2 + (1 - \frac{m+n}{p})\|w_1^*\|^2 + \frac{(m+n)\sigma^2}{p - (m+n) - 1}$$

and

$$\mathbb{E}[\mathcal{L}_2(w_{\vec{2}})] = \frac{n}{p}(\frac{m}{n} + \frac{m}{(p - (m+n) - 1)})\|w_2^* - w_1^*\|^2 + (1 - \frac{m+n}{p})\|w_2^*\|^2 + \frac{(m+n)\sigma^2}{p - (m+n) - 1}$$

Therefore, the generalization error is

$$\mathbb{E}[G(w_{\vec{2}})] = \frac{1}{2}(\mathbb{E}[\mathcal{L}_1(w_{\vec{2}})] + \mathbb{E}[\mathcal{L}_2(w_{\vec{2}})])$$

$$= \frac{n}{2p}(1 + \frac{m}{n} + \frac{2m}{(p - (m+n) - 1)})\|w_2^* - w_1^*\|^2 + (1 - \frac{m+n}{p})\|w_1^*\|^2 + \frac{(m+n)\sigma^2}{p - (m+n) - 1}$$

Similarly,

$$\mathbb{E}[\mathcal{L}_1(w_{\vec{1}})] = (1 - \frac{n}{p})\|w_1^*\|^2 + \frac{n\sigma^2}{p - n - 1}$$

Therefore,

$$\mathbb{E}[F(w_{\vec{2}})] = \mathbb{E}[\mathcal{L}_1(w_{\vec{2}})] - \mathbb{E}[\mathcal{L}_1(w_{\vec{1}})]$$

$$= \frac{n}{p}(1 + \frac{m}{(p - (m+n) - 1)})\|w_1^* - w_2^*\|^2 - \frac{m}{p}\|w_1^*\|^2 + \frac{(m+n)\sigma^2}{p - (m+n) - 1} - \frac{n\sigma^2}{p - n - 1}$$

$\square$

## A.4 Proof of Theorem 4.2

We start by proving a theorem for the general case:

**Theorem A.9.** *Assume $m_1 = m_2 = ... = m_t = m$ and $n_{t+1} = n$. Also denote $\bar{n}_{t+1} = tm + n$ and let $\alpha_t = 1 - \frac{n}{p-tm}$. Considering the continual learner described in Equation 16 when $p \geq \bar{n}_t + 2$, the loss of task $i$ at timestep $t$, can be recursively calculated as:*

$$\mathbb{E}[\mathcal{L}_i(w_{\overrightarrow{t+1}})] = \mathbb{E}[\|w_{\overrightarrow{t+1}} - w_i^*\|^2]$$

$$= \frac{m}{p}(1-\alpha_t) \sum_{s=1}^{t} \|w_s^* - w_i^*\|^2 + \frac{n}{p}\|w_{t+1}^* - w_i^*\|^2 + \frac{mn}{p(p-\bar{n}_t-1)} \sum_{s=1}^{t} \|w_{t+1}^* - w_s^*\|^2$$

$$+ \frac{m^2}{2p}\left(\frac{1}{p-\bar{n}_t-1} - \frac{\alpha_t}{p-(\bar{n}_t-n)-1}\right) \sum_{s=1}^{t}\sum_{s'=1}^{t} \|w_s^* - w_{s'}^*\|^2$$

$$+ \frac{\bar{n}_t\sigma^2}{p-\bar{n}_t-1} - \frac{\alpha_t(\bar{n}_t-n)\sigma^2}{p-(\bar{n}_t-n)-1}$$

$$+ \alpha_t\mathbb{E}[\|w_{\overrightarrow{t}} - w_i^*\|^2] \tag{23}$$

*Proof.* For notation simplicity, denote $r := t+1$. Using Lemma A.6 and the notation introduced in Section A.1, we know that:

$$w_{\overrightarrow{t+1}} = (I - P_{1:r})w_{\overrightarrow{t}} + X_{1:r}^\dagger y_{1:r}$$

where $P_{1:r} = X_{1:r}(X_{1:r}^\top X_{1:r})^{-1}X_{1:r}^\top$ and $X_{1:r}^\dagger = X_{1:r}(X_{1:r}^\top X_{1:r})^{-1}$. Notice that $X_{1:r} \in \mathbb{R}^{p\times(tm+n)}$ is the result of the concatenation of all data points in the memory to the new data for task $t+1$. Importantly, $X_{1:r}$ contains exactly $m$ columns from task $t$. Next we can write:

$$w_{\overrightarrow{t+1}} = (I - P_{1:r})w_{\overrightarrow{t}} + \sum_{s=1}^{r} P_{0|s}w_s^* + X_{1:r}^\dagger z_{1:r}$$

where $P_{0|s} = X_{1:r}(X_{1:r}^\top X_{1:r})^{-1}X_{0|s}$. Therefore,

$$\mathbb{E}[\mathcal{L}_i(w_{\overrightarrow{t+1}})] = \mathbb{E}[\|w_{\overrightarrow{t+1}} - w_i^*\|^2]$$

$$= \mathbb{E}[\|\sum_{s=1}^{r} P_{0|s}(w_s^* - w_i^*) + (I - P_{1:r})(w_{\overrightarrow{t}} - w_i^*) + X_{1:r}^\dagger z_{1:r}\|^2] \qquad (\sum_{s=1}^{r} P_{0|s} = P_{1:r})$$

$$= \mathbb{E}[\|\sum_{s=1}^{r} P_{0|s}(w_s^* - w_i^*) + (I - P_{1:r})(w_{\overrightarrow{t}} - w_i^*)\|^2] + \mathbb{E}[\|X_{1:r}^\dagger z_{1:r}\|^2] \qquad (z_{1:r} \perp\!\!\!\perp X_{1:r}, \mathbb{E}[z_{1:r}] = 0)$$

$$= \mathbb{E}[\|\sum_{s=1}^{r} P_{0|s}(w_s^* - w_i^*)\|^2] + \mathbb{E}[\|(I - P_{1:r})(w_{\overrightarrow{t}} - w_i^*)\|^2] + \mathbb{E}[\|X_{1:r}^\dagger z_{1:r}\|^2] \qquad ((I - P_{1:r})P_{0|s} = 0)$$

The form of this expression is very similar to what we used in the proof of Theorem 3.2 in Section A.2.2. The only considerable difference that causes complications is the term $\mathbb{E}[\|(I - P_{1:r})(w_{\overrightarrow{t}} - w_i^*)\|^2]$. The reason is that the term $w_{\overrightarrow{t}}$ is no longer independent of $P_{1:r}$, and it is not straightforward to calculate this expectation. Therefore, we avoid repeating the other steps and only focus on finding this term:

$$\mathbb{E}[\|(I - P_{1:r})(w_{\overrightarrow{t}} - w_i^*)\|^2] = \mathbb{E}[\|w_{\overrightarrow{t}} - w_i^*\|^2] - \mathbb{E}[\|P_{1:r}(w_{\overrightarrow{t}} - w_i^*)\|^2]$$

Notice that $w_{\overrightarrow{t}}$ is independent of $X_r$. Therefore, we decompose the second term using Lemma A.3 part (i):

$$\mathbb{E}[\|P_{1:r}(w_{\overrightarrow{t}} - w_i^*)\|^2] = \mathbb{E}[\|P_{1:t}(w_{\overrightarrow{t}} - w_i^*) + (I - P_{1:t})X_r(X_r^\top(I - P_{1:t})X_r)^{-1}X_r^\top(I - P_{1:t})(w_{\overrightarrow{t}} - w_i^*)\|^2]$$

$$= \mathbb{E}[\|P_{1:t}(w_{\overrightarrow{t}} - w_i^*)\|^2] + \mathbb{E}[\|(I - P_{1:t})X_r(X_r^\top(I - P_{1:t})X_r)^{-1}X_r^\top(I - P_{1:t})(w_{\overrightarrow{t}} - w_i^*)\|^2]$$

Recall that $X_r \in \mathbb{R}^{p \times n}$ follows the standard normal distribution and is independent of $(w_{\overrightarrow{t}} - w_i^*)$ and $P_{1:t}$. Due to the rotational invariance property, we can write:

$$\mathbb{E}[\|P_{1:r}(w_{\overrightarrow{t}} - w_i^*)\|^2] = \mathbb{E}[\|P_{1:t}(w_{\overrightarrow{t}} - w_i^*)\|^2] + (\frac{n}{p - tm})\mathbb{E}[\|(I - P_{1:t})(w_{\overrightarrow{t}} - w_i^*)\|^2]$$

$$= (1 - \frac{n}{p - tm})\mathbb{E}[\|P_{1:t}(w_{\overrightarrow{t}} - w_i^*)\|^2] + (\frac{n}{p - tm})\mathbb{E}[\|w_{\overrightarrow{t}} - w_i^*\|^2]$$

Therefore, it suffices to focus on $\mathbb{E}[\|P_{1:t}(w_{\overrightarrow{t}} - w_i^*)\|^2]$ as the next step. We go back to the definition of $w_{\overrightarrow{t}}$ and use Lemma A.6 again. Notice that $w_{\overrightarrow{t}}$ was trained on the samples in the memory from tasks 1 to $t - 1$ (each task with size $m$) in addition to the data from task $t$ with size $n$. Let's build a new matrix $\hat{X}_{1:t} \in \mathbb{R}^{p \times ((t-1)m+n)}$ to represent all of the training data used in that step. Similarly, build $\hat{y}_{1:t}$ and $\hat{z}_{1:t}$ by concatenating all the label and noise values. Also denote $\hat{X}_{0|s} \in \mathbb{R}^{p \times ((t-1)m+n)}$ with zeros at all columns except the columns corresponding to task $s$. Then we can write:

$$w_{\overrightarrow{t}} = (I - \hat{P}_{1:t})w_{\overrightarrow{t-1}} + \sum_{s=1}^{t} \hat{P}_{0|s}w_s^* + \hat{X}_{1:t}^\dagger \hat{z}_{1:t}$$

where $\hat{X}_{1:t}^\dagger = \hat{X}_{1:t}(\hat{X}_{1:t}^\top \hat{X}_{1:t})^{-1}$, $\hat{P}_{1:t} = \hat{X}_{1:t}(\hat{X}_{1:t}^\top \hat{X}_{1:t})^{-1}\hat{X}_{1:t}^\top$ and $\hat{P}_{0|s} = \hat{X}_{1:t}(\hat{X}_{1:t}^\top \hat{X}_{1:t})^{-1}\hat{X}_{0|s}^\top$. Notice that the first $tm$ columns of $\hat{X}_{1:t}$ is exactly $X_{1:t}$ and there are exactly $n - m$ extra columns in $\hat{X}_{1:t}$ that we threw away before proceeding to the task $t + 1$. Let $\hat{X}_t \in \mathbb{R}^{p \times (n-m)}$ represent this portion of the training set. In other words, $\hat{X}_{1:t} = [X_{1:t} \quad \hat{X}_t]$. With this in mind, based on Lemma A.3 part (v), it holds that

$$\hat{X}_{1:t}^\dagger = [X_{1:t}^\dagger - (I - P_{1:t})\hat{X}_t(\hat{X}_t^\top(I - P_{1:t})\hat{X}_t)^{-1}\hat{X}_t^\top X_{1:t}^\dagger \quad (I - P_{1:t})\hat{X}_t(\hat{X}_t^\top(I - P_{1:t})\hat{X}_t)^{-1}]$$

$$\Rightarrow P_{1:t}\hat{P}_{1:t} = P_{1:t}\hat{X}_{1:t}^\dagger \hat{X}_{1:t}^\top = [X_{1:t}^\dagger \quad 0]\hat{X}_{1:t}^\top = P_{1:t}$$

and also

$$P_{1:t}\hat{P}_{0|s} = P_{1:t}\hat{X}_{1:t}^\dagger \hat{X}_{0|s}^\top = P_{0|s}$$

Therefore,

$$\mathbb{E}[\|P_{1:t}(w_{\overrightarrow{t}} - w_i^*)\|^2] = \mathbb{E}[\|P_{1:t}(I - \hat{P}_{1:t})(w_{\overrightarrow{t-1}} - w_i^*) + \sum_{s=1}^{t} P_{1:t}\hat{P}_{0|s}(w_s^* - w_i^*) + P_{1:t}\hat{X}_{1:t}^\dagger \hat{z}_{1:t}\|^2]$$

$$= \mathbb{E}[\|\sum_{s=1}^{t} P_{0|s}(w_s^* - w_i^*) + X_{1:t}^\dagger z_{1:t}\|^2]$$

We have previously calculated terms like this in Section A.2.2. Therefore, we can write overall:

$$\mathbb{E}[\|(I - P_{1:r})(w_{\vec{t}} - w_i^*)\|^2] = \mathbb{E}[\|w_{\vec{t}} - w_i^*\|^2] - (1 - \frac{n}{p - tm})\mathbb{E}[\|P_{1:t}(w_{\vec{t}} - w_i^*)\|^2] - (\frac{n}{p - tm})\mathbb{E}[\|w_{\vec{t}} - w_i^*\|^2]$$

$$= \alpha_t \mathbb{E}[\|w_{\vec{t}} - w_i^*\|^2] - \alpha_t \mathbb{E}[\|P_{1:t}(w_{\vec{t}} - w_i^*)\|^2]$$

$$= \alpha_t \mathbb{E}[\|w_{\vec{t}} - w_i^*\|^2] - \alpha_t [\frac{m^2}{2p(p - tm - 1)} \sum_{s=1}^{t} \sum_{s'=1}^{t} \|w_s^* - w_{s'}^*\|^2$$

$$+ \frac{m}{p} \sum_{s=1}^{t} \|w_s^* - w_i^*\|^2 + \frac{tm\sigma^2}{p - tm - 1}]$$

Finally, we have

$$\mathbb{E}[\mathcal{L}_i(w_{\overrightarrow{t+1}})] = \mathbb{E}[\|w_{\overrightarrow{t+1}} - w_i^*\|^2]$$

$$= \frac{m^2}{2p(p - \bar{n}_t - 1)} \sum_{s=1}^{t} \sum_{s'=1}^{t} \|w_s^* - w_{s'}^*\|^2 + \frac{mn}{p(p - \bar{n}_t - 1)} \sum_{s=1}^{t} \|w_{t+1}^* - w_s^*\|^2$$

$$+ \frac{m}{p} \sum_{s=1}^{t} \|w_s^* - w_i^*\|^2 + \frac{n}{p} \|w_{t+1}^* - w_i^*\|^2 + \mathbb{E}[\|(I - P_{1:r})(w_{\vec{t}} - w_i^*)\|^2] + \frac{\bar{n}_t \sigma^2}{p - \bar{n}_t - 1}$$

$$= \frac{m}{p}(1 - \alpha_t) \sum_{s=1}^{t} \|w_s^* - w_i^*\|^2 + \frac{n}{p} \|w_{t+1}^* - w_i^*\|^2 + \frac{mn}{p(p - \bar{n}_t - 1)} \sum_{s=1}^{t} \|w_{t+1}^* - w_s^*\|^2$$

$$+ \frac{m^2}{2p}(\frac{1}{p - \bar{n}_t - 1} - \frac{\alpha_t}{p - (\bar{n}_t - n) - 1}) \sum_{s=1}^{t} \sum_{s'=1}^{t} \|w_s^* - w_{s'}^*\|^2$$

$$+ \frac{\bar{n}_t \sigma^2}{p - \bar{n}_t - 1} - \frac{\alpha_t(\bar{n}_t - n)\sigma^2}{p - (\bar{n}_t - n) - 1}$$

$$+ \alpha_t \mathbb{E}[\|w_{\vec{t}} - w_i^*\|^2]$$

$$\square$$

Using the recursive form in Equation 23, one can exactly find the loss of the continual learner in Equation 16. However, the full form is not intuitive. Therefore, we focus on the two-task case instead.

**Proof of Theorem 4.2**

*Proof.* We use Equation 23 and substitute $\sigma = 0$ and $T = 2$. Therefore, $\alpha_0 = 1 - \frac{n}{p}$ and $\alpha_1 = 1 - \frac{n}{p - m}$. For the loss of the first task, we have

$$\mathbb{E}[\mathcal{L}_1(w_{\vec{1}})] = \mathbb{E}[\|w_{\vec{1}} - w_1^*\|^2] = (1 - \frac{n}{p})\|w_1^*\|^2$$

$$\Rightarrow \mathbb{E}[\mathcal{L}_1(w_{\overrightarrow{2}})] = \mathbb{E}[\|w_{\overrightarrow{2}} - w_1^*\|^2]$$

$$= \frac{n}{p}(1 + \frac{m}{p - (n+m) - 1})\|w_2^* - w_1^*\|^2 + (1 - \frac{n}{p-m})\mathbb{E}[\|w_{\overrightarrow{1}} - w_1^*\|^2]$$

$$= \frac{n}{p}(1 + \frac{m}{p - (n+m) - 1})\|w_2^* - w_1^*\|^2 + (1 - \frac{n}{p-m})(1 - \frac{n}{p})\|w_1^*\|^2$$

And for the second task:

$$\mathbb{E}[\mathcal{L}_2(w_{\overrightarrow{1}})] = \mathbb{E}[\|w_{\overrightarrow{1}} - w_2^*\|^2]$$

$$= \frac{n}{p}\|w_1^* - w_2^*\|^2 + (1 - \frac{n}{p})\|w_2^*\|^2$$

$$\Rightarrow \mathbb{E}[\mathcal{L}_2(w_{\overrightarrow{2}})] = \mathbb{E}[\|w_{\overrightarrow{t+1}} - w_i^*\|^2]$$

$$= \frac{m}{p}(\frac{n}{p-m} + \frac{n}{p - (m+n) - 1})\|w_1^* - w_2^*\|^2 + (1 - \frac{n}{p-m})\mathbb{E}[\|w_{\overrightarrow{1}} - w_2^*\|^2]$$

$$= \frac{n}{p}(1 - \frac{n-m}{p-m} + \frac{m}{p - (m+n) - 1})\|w_1^* - w_2^*\|^2 + (1 - \frac{n}{p-m})(1 - \frac{n}{p})\|w_2^*\|^2$$

Finally, we can find the desired metrics:

$$\mathbb{E}[G(w_{\overrightarrow{2}})] = \frac{1}{2}(\mathbb{E}[\mathcal{L}_1(w_{\overrightarrow{2}})] + \mathbb{E}[\mathcal{L}_2(w_{\overrightarrow{2}})])$$

$$= \frac{n}{2p}(2 - \frac{n-m}{p-m} + \frac{2m}{(p - \bar{n}_t - 1)})\|w_1^* - w_2^*\|^2 + (1 - \frac{n}{p-m})(1 - \frac{n}{p})\|w_1^*\|^2$$

and

$$\mathbb{E}[F(w_{\overrightarrow{2}})] = \mathbb{E}[\mathcal{L}_1(w_{\overrightarrow{2}})] - \mathbb{E}[\mathcal{L}_1(w_{\overrightarrow{1}})]$$

$$= \frac{n}{p}(1 + \frac{m}{p - (n+m) - 1})\|w_2^* - w_1^*\|^2 - \frac{n}{p-m}(1 - \frac{n}{p})\|w_1^*\|^2$$

$\square$

# B   Bridging Linear Models and Overparameterized DNNs

Recent literature has pointed out several connections between linear models and overparameterized DNNs. In this section, we briefly review this connection to highlight the importance of studying linear models and refer the reader to Section 1.2 of Hastie et al. (2022) for a broader discussion. We establish the connection using the concept of lazy training regime.

Assume an i.i.d. data as $\mathcal{D} = \{(z_i, y_i)\}_{i=1}^n$ with inputs $z_i \in \mathbb{R}^d$ and labels $y_i \in \mathbb{R}$. Consider a possibly non-linear neural network $f(\cdot; \theta) : \mathbb{R}^d \to \mathbb{R}$ parameterized by $\theta \in \mathbb{R}^p$. Under certain conditions (Jacot et al., 2018; Allen-Zhu et al., 2019; Chizat et al., 2020), including overparameterization, the neural network $f(\cdot; \theta)$ can be approximated by its first-order Taylor expansion around the initial parameters $\theta_0$. Furthermore, by supposing that the initialization is such that $f(z; \theta_0) \approx 0$, we can write the following linear form:

$$f(z; \theta) \approx f(z; \theta_0) + \nabla_\theta f(z; \theta_0)^\top (\theta - \theta_0) \approx \nabla_\theta f(z; \theta_0)^\top (\theta - \theta_0).$$

This approximation is still non-linear in $z$ but linear in the parameters $\theta$, and it implies that in the lazy training regime, the neural network behaves similarly to a linear model where the features are given by the gradients of the network with respect to its parameters. Specifically, the features are the Jacobian matrix $x_i := \nabla_\theta f(z_i; \theta_0)$.

Additionally, this approximation allows us to define a Neural Tangent Kernel (NTK) (Jacot et al., 2018) as $\Theta(z_i, z_j) = \nabla_\theta f(z_i; \theta_0)^\top \nabla_\theta f(z_j; \theta_0)$, which captures the inner product of the gradients of the neural network with respect to the parameters at different data points. As training progresses, if the parameters remain close to their initialization (lazy training), the predictions of the neural network can be effectively described by a linear model in this high-dimensional feature space defined by the NTK. This connection elucidates how overparameterized deep neural networks can exhibit behavior akin to kernel methods, where the NTK serves as the kernel function that determines the similarity between data points.

## C Experimental Details

We presented the most important details regarding the dataset and architecture details in Section 5. We only present extra information here.

We used the standard cross-entropy loss for all our models. For the continual learning models, we used a simple experience replay with a uniform sampling strategy in which we optimized the model simultaneously on the data for the new task and the previously stored samples in the memory buffer.

We utilized SGD optimizer with a learning rate of 0.01, Nestrov, and 0.95 momentum. We trained all our models for 100 epochs. Notice that there exists an epoch-wise double-descent in deep neural networks (Nakkiran et al., 2021). Therefore, we fixed the number of epochs to 100 and mostly explored the model-wise double descent in our work. For simplicity, no extra augmentation or preprocessing was applied to the dataset. All experiments in this paper were repeated at least 3 times, and the average results were reported.

**Further Details on Table 1** In Tables 2 to 5, we provide the full version of Table 1 with extra details for all four datasets. Notice that in the CIFAR-100 experiments, we used the frozen ResNet-18 backbone and used the width of the MLP layer to control the architecture budget. Similarly, for IN-R and CUB datasets we used a pretrained ViT-B-16 backbone and changed the model's complexity by modyfing the final MLP layers. However, for the PMNIST dataset, an unfrozen ResNet-18 was used and the whole backbone was optimized.

Table 2: The effect of varying sample memory budget and architecture memory budget over the average accuracy and average forgetting of a continual learner on the CIFAR-100 dataset. $k$ denotes the width of the MLP layers and $m$ is the number of samples in the memory for each class. The last task train error for all experiments is zero, meaning that all models operate in the overparameterized regime.

| $m$ | Sample Budget (MB) | $k$ | Arch. Budget (MB) | Avgerage Accuracy (%) | Avgerage Forgetting. (%) |
|---|---|---|---|---|---|
| 50 | 122.8 | 128 | 0.4 | $56.6_{\pm 0.4}$ | $-22.5_{\pm 0.5}$ |
| 40 | 98.3 | 1605 | 24.5 | $70.6_{\pm 0.1}$ | $-7.1_{\pm 0.1}$ |
| 30 | 73.7 | 2329 | 49.1 | $\mathbf{71.9_{\pm 0.2}}$ | $\mathbf{-5.4_{\pm 0.3}}$ |
| 20 | 49.1 | 2885 | 73.7 | $66.3_{\pm 0.4}$ | $-11.3_{\pm 0.4}$ |
| 10 | 24.5 | 3354 | 98.3 | $67.4_{\pm 0.3}$ | $-9.7_{\pm 0.3}$ |
| 0 | 0.0 | 3768 | 122.9 | $63.8_{\pm 0.3}$ | $-10.8_{\pm 0.3}$ |

Table 3: The effect of varying sample memory budget and architecture memory budget over the average accuracy and average forgetting of a continual learner on the IN-R dataset. $k$ denotes the width of the MLP layers and $m$ is the number of samples in the memory for each class. The last task train error for all experiments is zero, meaning that all models operate in the overparameterized regime.

| $m$ | Sample Budget (MB) | $k$ | Arch. Budget (MB) | Avgerage Accuracy (%) | Avgerage Forgetting. (%) |
|---|---|---|---|---|---|
| 10 | 1204.2 | 32 | 0.1 | $76.1_{\pm 0.4}$ | $-3.3_{\pm 0.6}$ |
| 8 | 963.4 | 5249 | 240.9 | $76.9_{\pm 0.6}$ | $-2.0_{\pm 0.3}$ |
| 6 | 722.5 | 7520 | 481.8 | $77.1_{\pm 0.6}$ | $-1.4_{\pm 0.4}$ |
| 4 | 481.7 | 9263 | 722.6 | $\mathbf{78.9_{\pm 0.5}}$ | $-1.0_{\pm 0.1}$ |
| 2 | 240.8 | 10733 | 963.5 | $78.5_{\pm 0.3}$ | $\mathbf{-0.6_{\pm 0.1}}$ |
| 0 | 0.0 | 12028 | 1204.4 | $49.5_{\pm 0.6}$ | $-20.5_{\pm 0.5}$ |

Table 4: The effect of varying sample memory budget and architecture memory budget over the average accuracy and average forgetting of a continual learner on the CUB dataset. $k$ denotes the width of the MLP layers and $m$ is the number of samples in the memory for each class. The last task train error for all experiments is zero, meaning that all models operate in the overparameterized regime.

| $m$ | Sample Budget (MB) | $k$ | Arch. Budget (MB) | Avgerage Accuracy (%) | Avgerage Forgetting. (%) |
|---|---|---|---|---|---|
| 5 | 602.1 | 16 | 0.1 | $87.6_{\pm 0.2}$ | $-4.5_{\pm 0.2}$ |
| 4 | 481.7 | 3644 | 120.4 | $84.1_{\pm 0.5}$ | $-6.7_{\pm 0.0}$ |
| 3 | 361.3 | 5249 | 240.9 | $88.5_{\pm 0.3}$ | $-1.8_{\pm 0.6}$ |
| 2 | 240.8 | 6481 | 361.4 | $\mathbf{92.0_{\pm 0.1}}$ | $\mathbf{-0.1_{\pm 0.2}}$ |
| 1 | 120.4 | 7520 | 481.8 | $91.7_{\pm 0.4}$ | $-0.4_{\pm 0.3}$ |
| 0 | 0.0 | 8435 | 602.1 | $75.8_{\pm 0.7}$ | $-12.2_{\pm 0.7}$ |

# D   Additional Experiments

In this section, we provide extra experiments on DNNs. Section D.1 provides the MTL experiments with single-head and multi-head architectures but with a frozen backbone. Section D.2 offers the results for the experiments with unfrozen backbones. In Section D.3, we dig deeper into investigating the optimal weights learned by each MLP layer, given that the layers are unshared across the tasks, and we measure the task similarity at each layer using the average inner product of the optimal task-specific weights. In Section D.4, we investigate the effect of the MLP depth and provide the corresponding train and test curves. Finally, in Section D.5 we offer extra experiments on CL models and demonstrate the effect of label noise on CL models.

## D.1   Complementary Multi-Task Learning Experiments

Figures 7 to 13 demonstrate the full training and testing curves for training the model in an MTL setup with multi-head and single-head architectures with different backbones and datasets. As discussed before in the main text, the training error is zero for both cases in large enough models. However, the multi-task learner performs worse in the single-head architecture due to the conflict at the classification layer.

## D.2   Multi-Task Learning Experiments with Unfrozen Backbone

Next, we perform similar experiments with an unfrozen backbone. The results of this experiment are provided in Figure 14. The model-wise double descent phenomena are similarly observable when training the backbone from scratch but at different locations for different configurations (refer to Section 5 for details on the architecture).

Table 5: The effect of varying sample memory budget and architecture memory budget over the average accuracy and average forgetting of a continual learner on the PMNIST dataset. $k$ denotes the width of the backbone layer, and $m$ is the number of samples in the memory for each class. The last task train error for all experiments is zero, meaning that all models operate in the overparameterized regime.

| $m$ | Sample Budget (MB) | $k$ | Arch. Budget (MB) | Avgerage Accuracy (%) | Avgerage Forgetting. (%) |
|---|---|---|---|---|---|
| 50 | 31.3 | 1 | 0.1 | $68.4_{\pm 1.6}$ | $-15.9_{\pm 0.2}$ |
| 40 | 25.0 | 24 | 6.4 | $83.2_{\pm 0.3}$ | $-11.4_{\pm 0.4}$ |
| 30 | 18.8 | 34 | 12.8 | $79.2_{\pm 1.0}$ | $-16.8_{\pm 1.1}$ |
| 20 | 12.5 | 42 | 19.5 | $\mathbf{86.3_{\pm 0.1}}$ | $\mathbf{-9.5_{\pm 0.1}}$ |
| 10 | 6.2 | 48 | 25.4 | $82.4_{\pm 0.3}$ | $-14.0_{\pm 0.4}$ |
| 0 | 0.0 | 54 | 32.1 | $30.8_{\pm 1.2}$ | $-71.5_{\pm 1.2}$ |

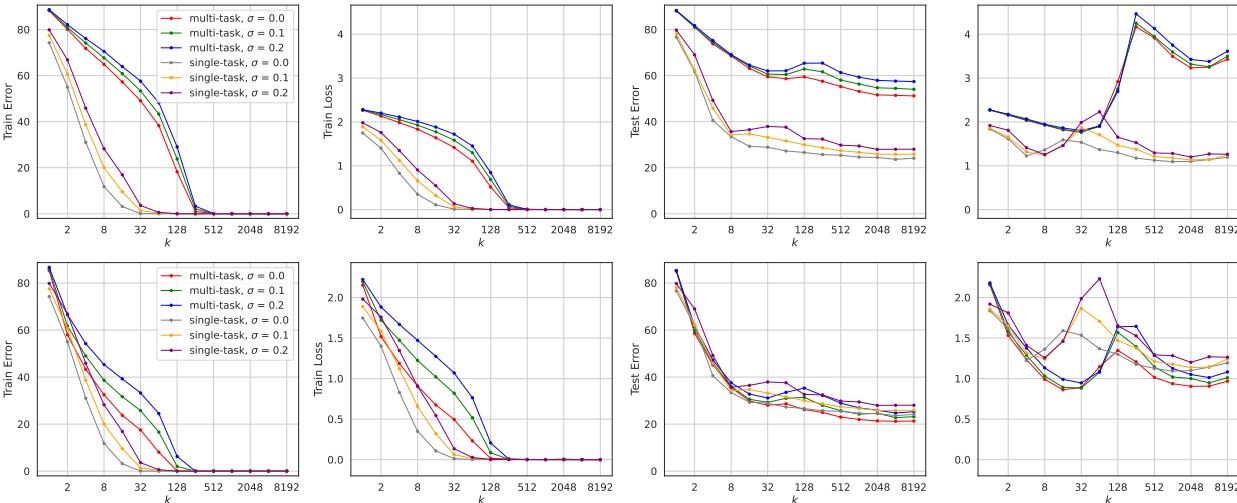

Figure 7: Comparing the single-task and multi-task performance on CIFAR-100 with pretrained ResNet-18 backbone. The top row shows the experiment with the single-head architecture and the bottom row corresponds to the multi-head architecture. The horizontal axis in all plots is the width of the MLP layer which is placed on top of the feature extractor. $\sigma$ represents the noisy portion of the training set that was corrupted by randomly switching its labels.

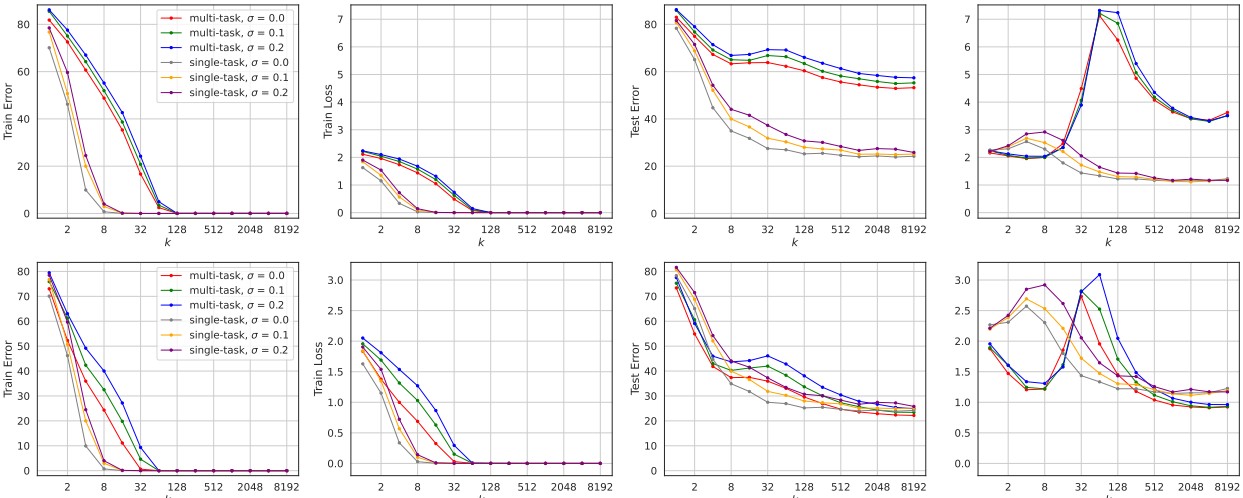

Figure 8: Comparing the single-task and multi-task performance on CIFAR-100 with pretrained ResNet-50 backbone. The top row shows the experiment with the single-head architecture, and the bottom row corresponds to the multi-head architecture.

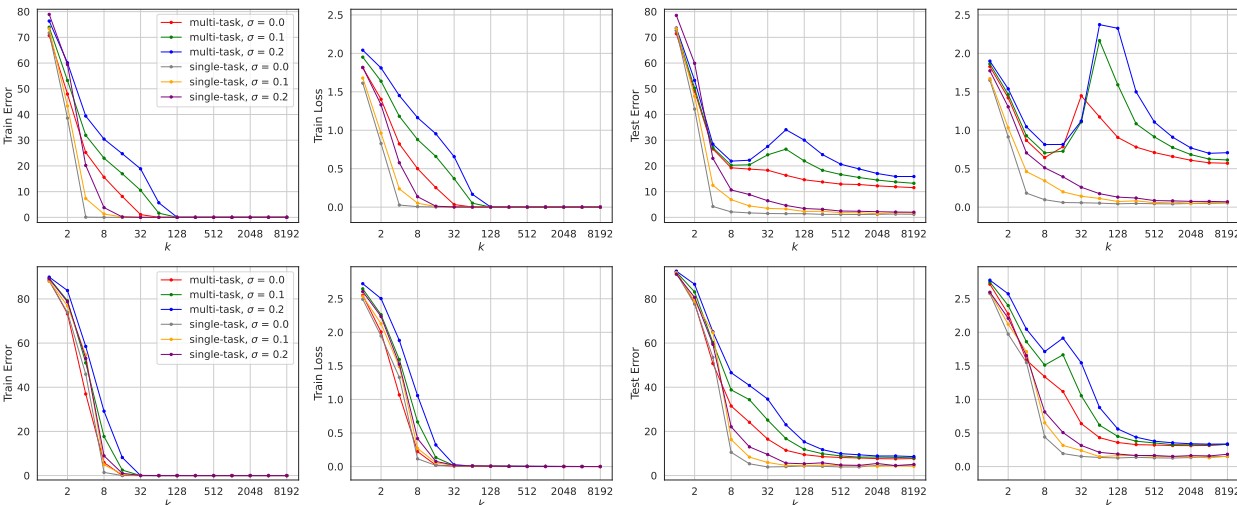

Figure 9: Comparing the single-task and multi-task performance on CIFAR-100 with pretrained ViT-B-16 backbone. The top row shows the experiment with the single-head architecture, and the bottom row corresponds to the multi-head architecture.

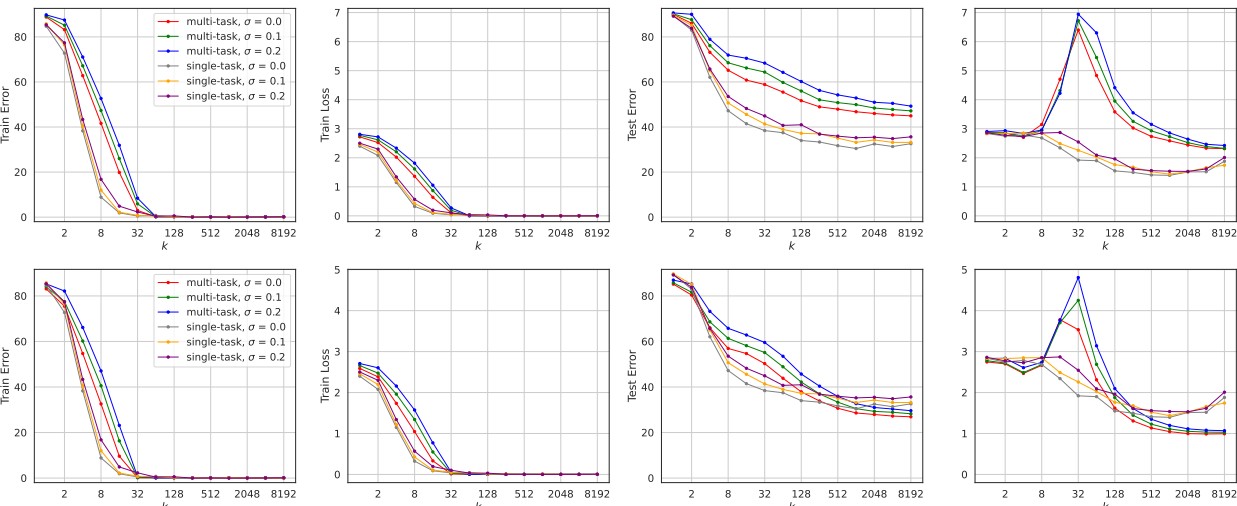

Figure 10: Comparing the single-task and multi-task performance on the Imagenet-R dataset with pretrained ResNet-50 backbone. The top row shows the experiment with the single-head architecture, and the bottom row corresponds to the multi-head architecture.

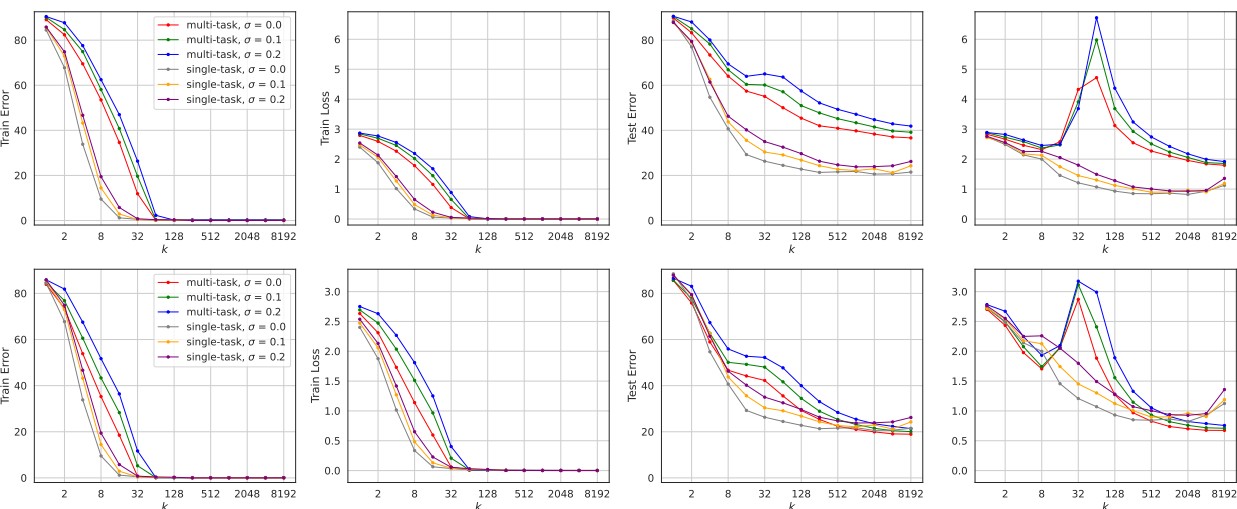

Figure 11: Comparing the single-task and multi-task performance on the Imagenet-R dataset with pretrained ViT-B-16 backbone. The top row shows the experiment with the single-head architecture, and the bottom row corresponds to the multi-head architecture.

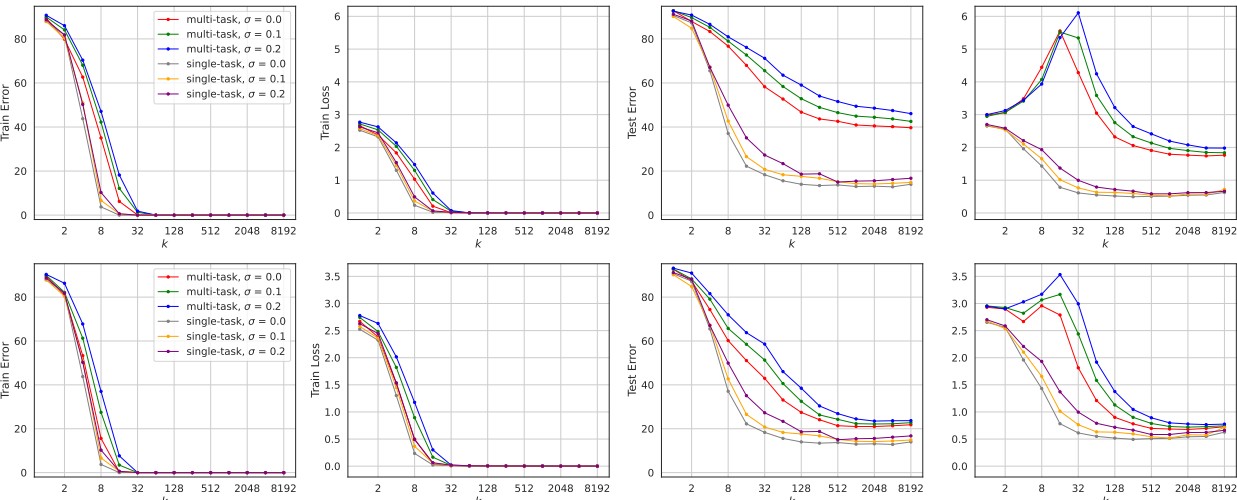

Figure 12: Comparing the single-task and multi-task performance on the CUB dataset with pretrained ResNet-50 backbone. The top row shows the experiment with the single-head architecture, and the bottom row corresponds to the multi-head architecture.

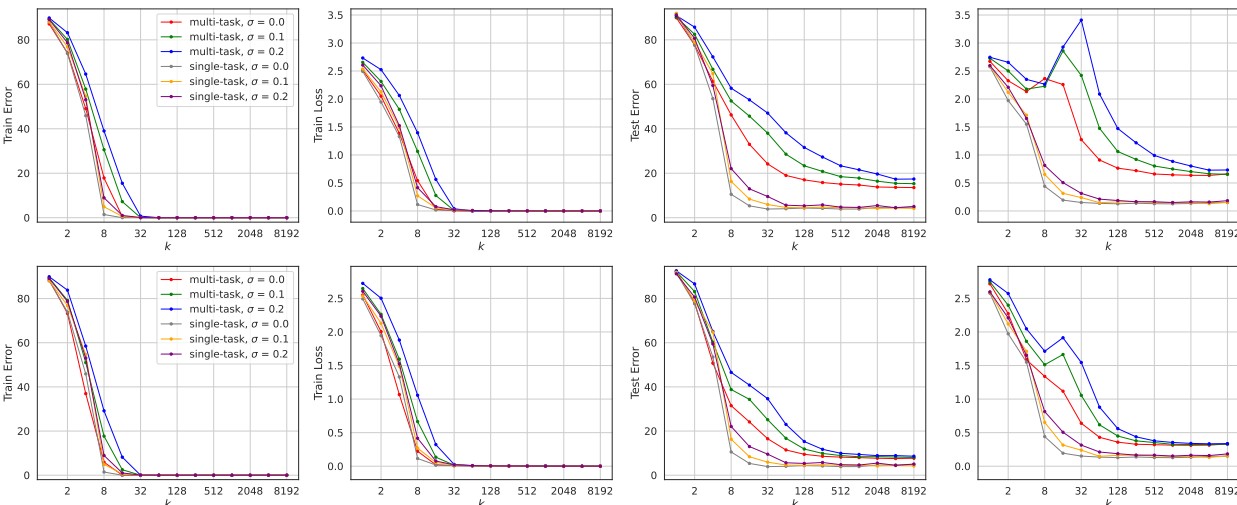

Figure 13: Comparing the single-task and multi-task performance on the CUB dataset with pretrained ViT-B-16 backbone. The top row shows the experiment with the single-head architecture, and the bottom row corresponds to the multi-head architecture.

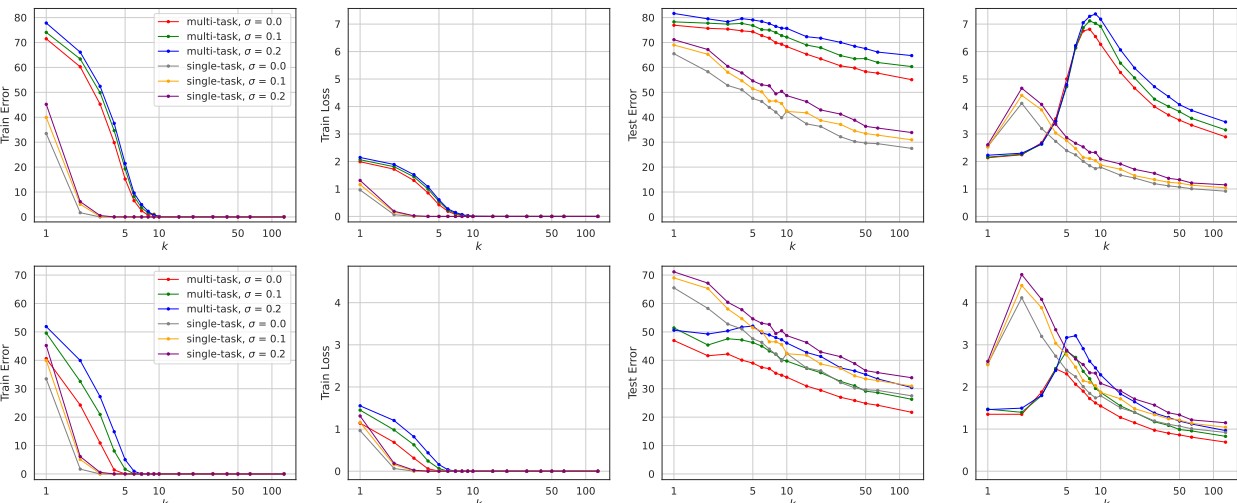

Figure 14: Comparing the single-task and multi-task performance on CIFAR-100 with unfrozen ResNet-18 backbone with random initialization. The top row shows the experiment with the single-head architecture, and the bottom row corresponds to the multi-head architecture. The horizontal axis in all plots is the scale factor that controls the number of filters in convolutional layers. $\sigma$ represents the noisy portion of the training set that was corrupted by randomly switching its labels.

### D.3 Investigation of the Task Optimal Weights

In this section, we analyze the layer-wise optimal weights for each training task. For this purpose, we start with a fully shared MTL model. Remember that in our MTL model, a frozen backbone is followed by three layers of MLP and a single-head classifier. At each time, we pick a specific layer and unshare it across the tasks by reinstantiating a new weight matrix for each task. Then, we optimize the model on the whole training set while ensuring that the unshared layers only see the data of their own task. We repeat this procedure by unsharing only one layer at a time to have task-specific optimal weights for each layer.

Figure 15 shows the results of this experiment. As demonstrated, the fully shared MTL model has the worst performance, suggesting that weight sharing does not happen effectively in a fully shared model. Although unsharing any layer can significantly boost performance, the experiments show that an unshared classification head (i.e., multi-head architecture) is the most effective.

To better understand the reason for such behavior, we look at the average inner product of flattened task-wise optimal weights. More formally, let $W_{t,l}^*$ represent the linearized weight matrix at layer $l$, optimized specifically for task $t$, where $1 \leq l \leq 4$ and $1 \leq t \leq T$. We compute the following layer-wise average score:

$$S_l = \frac{2}{T(T-1)} \sum_{t=1}^{T} \sum_{t'=t+1}^{T} \langle W_{t,l}^*, W_{t',l}^* \rangle. \tag{24}$$

Figure 16 shows the values of this score at different layers for different model sizes. Although the classification layer has less parameters than the other layers (because it is bonded to the number of classes from one side), the figure shows that it has the most negative score. In fact, the negative average score shows that the tasks are mostly conflicting at this layer and this explains why unsharing the final layer makes more improvements than the other layers.

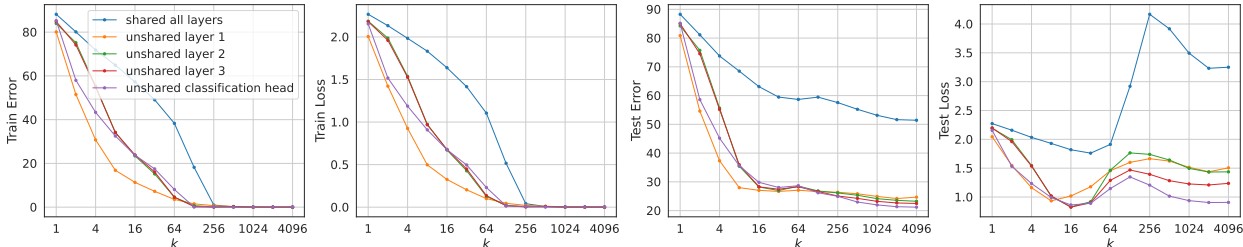

Figure 15: Investigating the effect of weight sharing in MTL by unsharing different layers of the MLP layer and training task-specific layers. The horizontal axis is the width of the MLP layer.

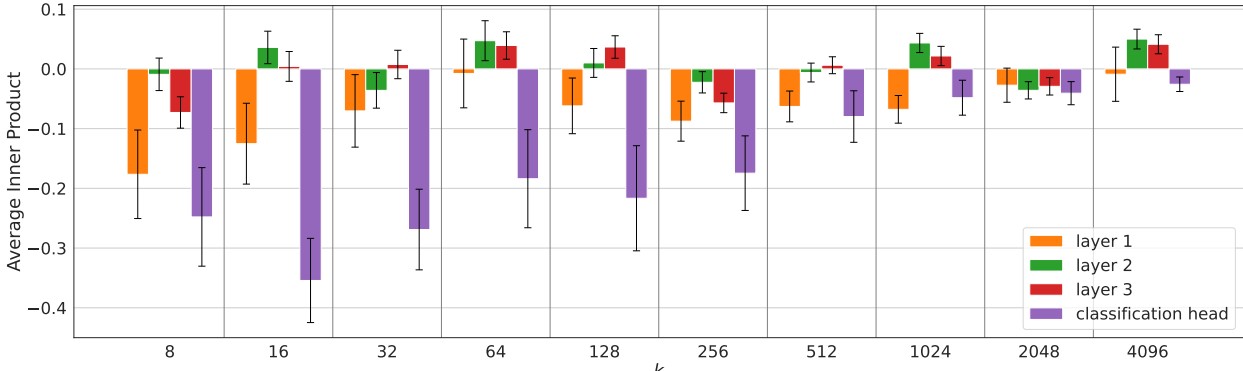

Figure 16: Comparing the average inner product score (Equation 24) across different layers after unsharing the weights across the tasks. The horizontal axis is the width of the MLP layer.

## D.4 Investigation of the Depth

In this section, we investigate the effect of the MLP depth on top of the pretrained backbones. Figure 17 compares different models with varying depths of the MLP. Notice that our theories are based on the assumption of a linear model, and therefore, they do not directly predict the behavior as a function of the depth. However, as discussed in Section B, the DNN performs similarly to a linear model as long as certain conditions hold. Figure 17 confirms this claim by showing similar behavior for all different models. Another interesting observation in this plot is the impact of depth over the height of the test error peak, which is a notable direction for future studies.

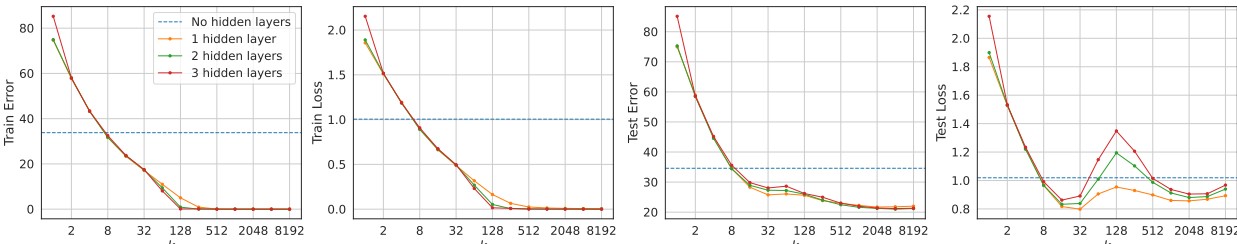

Figure 17: Visualizing the effect of the depth on the performance curves of the multi-head multi-task learner. The horizontal axis is the width of the MLP layers. Notice that a simple linear layer on top of the feature extractor can not fully overfit the training samples and has no tunable hyperparameter for increasing the complexity.

### D.5 Complementary Continual Learning Experiments

Figure 18 demonstrates full training and testing curves of a CL setup on CIFAR-100 with different replay buffer sizes using pretrained ResNet-18. Notice that the last task training error is zero for both cases in large enough models. Notice that Figure 18 contains both experiments with a single-head and multi-head architecture, which is respectively denoted as task-incremental and domain-incremental learning in the CL literature. Additionally, Figure 19 shows a similar experiment with an unfrozen ResNet-18 backbone.

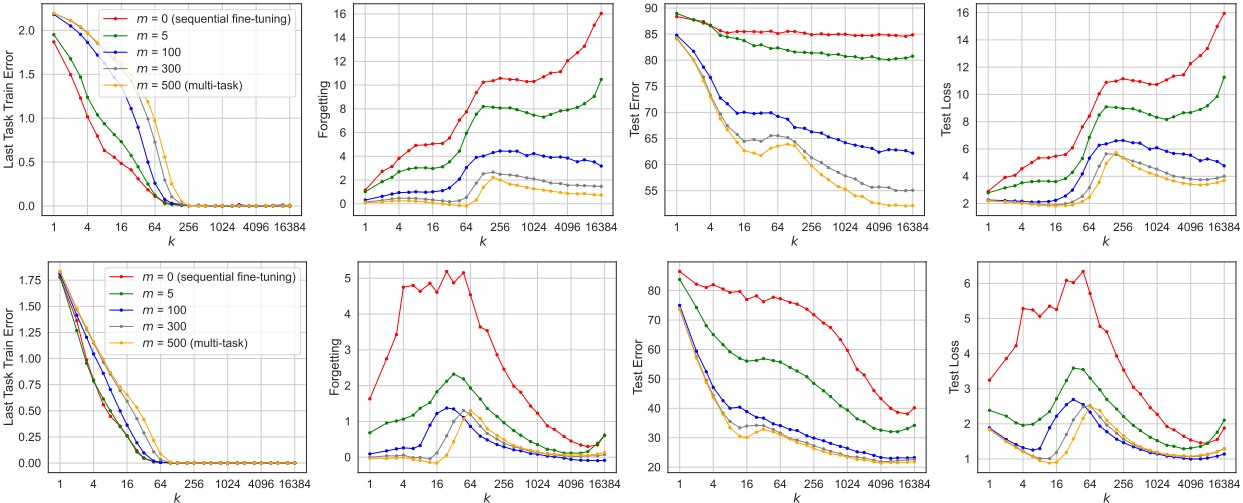

Figure 18: Visualizing the CL performance on CIFAR-100 with different values of memory budget denoted as $m$. The horizontal axis is the width of the MLP layer. The top row corresponds to the experiments with a single-head classifier and the bottom row shows the multi-head architecture.

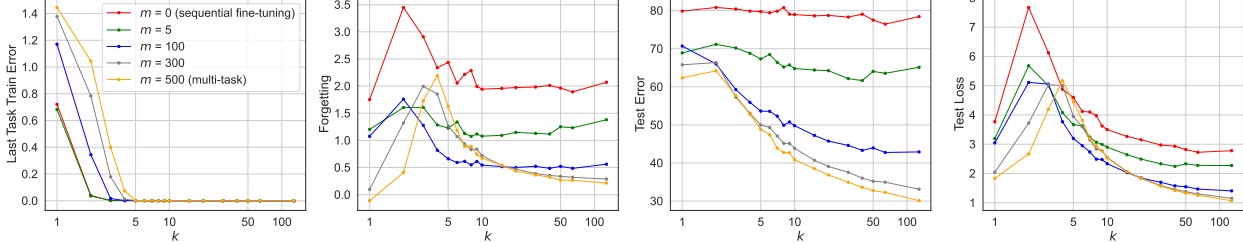

Figure 19: Visualizing the CL performance on CIFAR-100 using an unfrozen ResNet-18 backbone and a multi-head classification head. The plot shows the performance for different values of memory budget denoted as $m$. The horizontal axis is scale factor that controls the number of filters in the ResNet-18 backbone.

Finally, we present the experiments with different noise levels in Figure 20. As our theories suggest, the error peak is intensified with stronger noise.

## E Limitations and Future Work

In this paper, we presented a theoretical framework for understanding multi-task overparameterized linear models which is novel and unexplored before. The strength of our work is that we offer exact theoretical closed-form results rather than asymptotic results or generalization bounds for an MTL setup. However, there are also limitations in our work, which we will address in this section.

Our theoretical analysis is based on several assumptions and simplifications to derive closed-form expressions. For instance, we assume i.i.d. Gaussian features, the existence of optimal task vectors and additive noise

Figure 20: Visualizing the CL performance of multi-head models using memory size $m = 50$ on CIFAR-100 with different levels of noise denoted as $\sigma$. The horizontal axis is the width of the MLP layer.

in our linear regression setup, which simplifies the mathematical analysis but may not fully represent the complexity and variability of real-world data. These assumptions might limit the generalizability of our results to more complex or non-Gaussian data distributions commonly encountered in practical applications.

The next potential limitation is that our analysis is focused on linear models. Although we connected the linear models and DNNs in Appendix B, but the connection holds under certain assumptions which may be violated in different applications. Besides, we emphasize that the goal of this work is not to make formal claims regarding the performance of multi-task DNNs, rather, we tried to qualitatively connect our understanding of linear models to the observations derived experimentally from DNNs. We believe that knowing the trade-offs between different system specifications, such as the model size and number of training samples, can provide an important guideline when deploying practical models.

Despite all these limitations, we believe that our work should be evaluated in line with the recent advancements in theoretical investigations of overparameterized models. Additionally, we emphasize that the theoretical tools developed in the proofs section are valuable and novel if compared to the related prior theoretical works in the field.

At this time, our theoretical tools for fully understanding overparameterized DNNs are limited and therefore, one possible direction for future work is to focus more specifically on these models. In the context of multi-task learning, it is important to understand what does task similarity mean in deep models operating over complex datasets. Remember that we defined the task similarity using the distance between the optimal task vectors, an assumption which is not necessarily well-defined for DNNs with high-dimensional weights. Therefore, it is essential to provide a better definition of the task similarity and then connect it to the double descent behavior of DNNs.

## F  Infrastructures and Data Availability

We mainly used PyTorch (Paszke et al., 2019) for our implementations and datasets are freely accessible (Imagenet-R (Hendrycks et al., 2021), CUB(Wah et al., 2011), CIFAR-100 (Krizhevsky et al., 2009), and MNIST (Deng, 2012)). All experiments in the main text are reproducible on a single NVIDIA 2080 TI GeForce GPU.

