# OpenReview forum: "Theoretical Insights into Overparameterized Models in Multi-Task and Replay-Based Continual Learning"
_TMLR — Accepted by TMLR_

### Review · Reviewer_f14z · 2024-12-29

**Summary Of Contributions:**

This paper investigates Continual Learning (CL) and Multi-Task Learning (MTL) within the context of overparameterized linear models, developing theoretical results that describe how various system parameters influence generalization error and knowledge transfer. Empirical evaluations demonstrate that these theoretical findings apply to deep neural networks.

**Audience:**

Yes

**Broader Impact Concerns:**

No ethical concerns.

**Claims And Evidence:**

Yes

**Requested Changes:**

Overall, I have not thoroughly examined the theoretical results, as I am not an expert in this area.

**Strengths And Weaknesses:**

Strengths:

This paper provides a detailed and unified characterization of Multi-Task Learning (MTL) and Continual Learning (CL) under various settings using linear models. The paper is well written and organised, making it enjoyable to read.

Weaknesses:

1. It would be beneficial to conclude practical insights. For instance, stating that "larger models with greater capacity are less vulnerable to forgetting" is not particularly surprising.
2. In Equations (12) and (13), why must the equality constraints be strictly satisfied? I believe there may be some noise in the data.
3. Better to discuss some recent works, e.g. [1].

[1] Online Continual Learning via Logit Adjusted Softmax, TMLR'24

---

> ### Author Response · Authors · 2025-01-06
>
> We appreciate your time and are glad that you valued our work. Please read our general note first. Here are our responses to the raised concerns:
>
> **1- Practical insights:** Exact theoretical analysis of deeper networks is extremely challenging with the existing theoretical toolkit and works on these models are limited for this reason. However, the complex models behave similarly to simpler models under certain conditions (e.g. lazy training regime). As a result, many works in the literature study linear models as a proxy [1-5]. More importantly, our empirical experiments with deeper models show that our results can be used to explain similar behaviors in MTL setup. We provided several empirical investigations in Section 5.2 to show the applicability of our results to modern DNNs. Also, please refer to Appendix D for a comprehensive discussion on the connection between DNNs and linear models.
>
>
> [1] Surprises in High-Dimensional Ridgeless Least Squares Interpolation, Hastie et al.
>
> [2] Generalization Guarantees for Neural Networks via Harnessing the Low-rank Structure of the Jacobian, Oymak et al, ArXiv preprint 2019.
>
> [3] Neural Tangent Kernel: Convergence and Generalization in Neural Networks, Jacot et al., NeurIPS 2018.
>
> [4] Learning and generalization in overparameterized neural networks, going beyond two layers, Allen-Zhu et al., NeurIPS 2019.
>
> [5] On lazy training in differentiable programming, Chizat et al, NeurIPS 2018.
>
>
> **2- Overfitting to noise:** Thanks for bringing up this question. Overparametrized models (especially modern DNNs) have many more parameters than needed and, therefore, are capable of fully memorizing the training data [1]. In fact, SGD is strong enough to completely fit the training data with our overparametrized model. However, despite the (noisy) training data being fully overfitted, the model shows remarkable generalization to the unseen test samples. This surprising behavior is called benign overfitting in the literature [2,3] and is not specific to MTL or CL setups. In fact, previous studies show that SGD benefits from an implicit regularization, which mitigates the effect of noisy training data [4,5,6].
>
> [1] Understanding deep learning requires rethinking generalization, Zhang et al., ICLR 2017.
>
> [2] Benign Overfitting in Linear Regression,  Bartlett et al., 2020.
>
> [3] Benign, Tempered, or Catastrophic: A Taxonomy of Overfitting, Mallinar, 2022.
>
> [4] Surprises in High-Dimensional Ridgeless Least Squares Interpolation, Hastie et al.
>
> [5] Characterizing implicit bias in terms of optimization geometry, Gunasekar et al., ICML 2018.
>
> [6] Two models of double descent for weak features, Mikhail Belkin et al., SIAM Journal on Mathematics of Data Science
>
> **3- Connection to recent studies:** Thanks for bringing our attention to this work. The paper you mentioned follows a different setup and approach compared to ours. They use a probabilistic perspective to solve the intra-class imbalance problem, while we use an optimization perspective to study the regression setup. We will cite this paper and also will connect our work to more recent theoretical studies in continual learning. Please let us know if there are other works that we have missed. Please also refer to the related work presented in Section 2 of our paper, in which we drew connections to some recent theoretical CL and MTL papers.

---

> > ### Comment · Reviewer_f14z · 2025-02-23
> > **Thanks for the responses**
> >
> > Thanks to the authors for their detailed responses. I have read the responses and other reviews. My concerns are addressed.

---

> > > ### Author Response · Authors · 2025-02-23
> > > **Follow-Up with the reviewer**
> > >
> > > Dear Reviewer,
> > >
> > > Thank you for your feedback that helped us to improve our work. We appreciate your engagement and reading our response.
> > >
> > > Best,
> > >
> > > Our Team

---

### Review · Reviewer_5YYX · 2025-01-10

**Summary Of Contributions:**

- This paper develops a theoretical framework to analyze the generalization error and knowledge transfer in overparameterized linear models used for MTL and replay-based CL. It explores how model size, dataset size, and task similarity impact these metrics. Notably, it characterizes the interplay of these parameters in both collaborative and conflicting task scenarios.

- The study extends the theoretical analysis to replay-based continual learning models, focusing on how buffer size and model capacity influence forgetting rates and generalization error. This helps understand the effectiveness of memory buffer strategies.

- This paper conducts extensive experiments using both linear models and deep neural networks, on CIFAR, ImageNet-R, and CUB-200 datasets. The paper demonstrates that its theoretical insights generalize to more complex systems.

**Audience:**

Yes

**Claims And Evidence:**

Yes

**Requested Changes:**

- The work shows a peak exists in the generalization error curve of the multi-task learner when increasing model complexity. I found this interesting. It would be great if the peak could be explicitly explained after presenting theoretical results.
- There are many findings and implications. It would be great to clearly outline them after the introduction, and make them parallel in the theoretical results and experimental results.
- In Section 4, the paper mentioned that "the continual learner can sometimes outperform the MTL baseline even when tasks are collaborative, especially at the multi-task learner’s interpolation threshold." Is there any connection to the theoretical results?
- There are many experimental results. It would be great to provide a summary at the beginning of the experiment section.
- How does this paper define task similarity/dissimilarity? It would be great to provide a mathematical definition.

**Strengths And Weaknesses:**

### Strengths
- The paper offers theoretical analyses into overparameterized linear models in MTL and CL for understanding generalization and knowledge transfer.
- It validates its theories through experiments on real-world datasets and architectures, showing applicability to deep neural networks.
- The study also shows trade-offs in memory size and model capacity in continual learning.

### Weaknesses
- The novelty compared to existing generalization bound results is not explicitly stated. The paper claims that they derive the first generalization-bound results for MTL with linear and overparameterized models. How about simply applying existing bounds to MTL settings (for example Hastie et al. (2022) cited in the paper), and what advantage do the proposed results have over the naive ones? How do the results differ from Wu et al. (2020)?
- The implications of the theoretical results are not clear. How well does the bound track generalization errors in practice? and Do they suggest some algorithmic design based on the theoretical results?

---

> ### Author Response · Authors · 2025-01-19
>
> We thank the reviewer for their feedback and appreciate them for recognizing the strength of our work. Please read our general note first. We tried to address your concerns below:
>
> **Weaknesses**
>
> 1- Novelty and Comparison to existing frameworks: Existing frameworks are incapable of fully capturing our results. For example, Hastie et al. (2022) have no notion of “task”, and therefore, their analysis is not appropriate for the case where the data is coming from multiple sources. In fact, we offer a more generalized framework, and their Theorem 1 is indeed a special case of our Theorem 3.1 when there is only a single task for learning. Although our main text may not fully convey this message, the complexity of our proofs, even when building upon Hastie et al. (2022), is non-trivial. (Please also refer to the bottom of page 5 where we explicitly highlighted this difference as well as the second paragraph in the third part of the Related work).
>
> Compared to Wu et al. (2020), we are similar in the data generation process, but their most important result, which is Theorem 2 in the paper, differs from ours in many aspects. First, we consider the “overparameterized regime” where the model can fully overfit the training data, while in their studies, they confine the width of the model or equivalently, study the cases with data abundance. Moreover, we focus on the implicit bias and the convergence point of GD while they merely show the existence of an optimal solution and study its generalization properties. In their Theorem 2, they provide an “upper bound” on the error of such optimal solution, when having only T=2 tasks. Meanwhile, we derive an “exact characterization” of the error for the general T task settings in our Theorem 3.1. Therefore, we believe that our results are not explainable by their approach.
>
>
> 2- Implications of theoretical results: In our studies, we were concerned with a theoretical understanding of multi-task learning in overparameterized linear models, and for practical implications, we tried to connect linear models to DNNs through experimental observations. In summary, our most important observations are as follows:
>
> + MTL DNNs experience a test error peak at the interpolation threshold, but unlike STL models, the peak not only depends on the label noise but is also intensified when tasks are highly dissimilar or ineffective parameter sharing is happening (See Figures 2 and 3).
> + The distance of optimal task-specific parameters correlates with the amount of conflict that is happening across different tasks and can provide evidence for deciding which layers to decouple (Please see Figure 4 and Section G.3 in the Appendix).
> + Memory-based continual learners also undergo a double descent as the number of parameters is increased. The characteristics of the error peak depend on the memory size as well as task-similarity and training size.
> + High overparametrization can diminish the negative knowledge transfer (forgetting). Additionally, overparametrized models benefit less from explicit memory buffers. Therefore, it is important to adjust the resources accordingly between model parameters and raw training samples.

---

> > ### Author Response · Authors · 2025-01-19
> >
> > **Requested Changes**
> >
> > Thanks for your suggestions and the requested changes. We will wait until other reviewers publish their reviews and reupload a revised version afterward to reflect your comments. We will incorporate the requested changes as follows:
> >
> > (i) The error peak is not specific to MTL or CL, rather it is a generally observed phenomenon even in STL. Therefore, we only highlighted how MTL setup might affect the characteristics of the error peak in our work. In fact, in the underparameterized regime, the model doesn’t have enough capacity to learn all training samples. This serves as a regularization and results in generalization. In the overparameterized regime, the model has much more parameters and can fully memorize the training samples. However, the implicit regularization of SGD favors low-norm solutions and avoids catastrophic overfitting.
> >
> > The test error peak occurs exactly between the two regimes, i.e. at the interpolation threshold, where the number of parameters equals the number of training data points. At this point, there exists exactly one model that can perfectly fit the training data, achieving zero training error. However, small fluctuations or noise in the training data lead to large variations in the learned parameters, resulting in high variance. This sensitivity causes poor generalization, displaying a spike in test error.
> >
> > In the prior STL studies, the variance stems only from label noise $\sigma$. However, we showed that in an MTL setup, even if $\sigma=0$, we can see a high test error on average because of the misalignment between the tasks. In fact, the stochasticity of data points (i.e. $X$) together with high task dissimilarity can lead to highly variant predictors with poor generalization behavior.
> >
> > (ii) Thanks for the suggestion. We will certainly mention the list of implications and results more clearly at the beginning of each section. Please refer to the list of implications mentioned in the previous response.
> >
> > (iii) Our theoretical results fully support the mentioned behaviors. For example, to see the mentioned claim, consider the generalization error in Equation (14). Assume an extreme case where $m=0$ (i.e. the regularized continual learner with zero memory capacity), then
> >
> > $$G_{CL} = \frac{n}{p} (2 - \frac{n}{p}) \Vert w^*_1 - w^*_2 \Vert^2 + (1 - \frac{n}{p})^2 \Vert w^*_1\Vert^2$$
> >
> > Similarly, applying $T=2$, $\Vert w_1^* \Vert = \Vert w_2^* \Vert$ and $n_1 = n_2 = n$ to Equation (10), yields the following error for the MTL model:
> >
> > $$G_{MTL} = \frac{n}{p} (1 + \frac{n}{p-2n-1}) \Vert w^*_1 - w^*_2 \Vert^2 + (1 - \frac{2n}{p}) \Vert w^*_1\Vert^2$$
> >
> >
> > Setting $p \approx 2n + 1$ makes a very large MTL error, while this is not true for the CL model.
> >
> > (iv) Sure. Please refer to (ii).
> >
> > (v) Each task is identified by its generative optimal weight vectors. In page 6, observation 2, we implicitly defined task similarity by introducing the l2 distance between these weight vectors as a measure of similarity. We will make sure to mention that more explicitly in the revised version of the paper.
> >
> > Please let us know if the abovementioned modifications would satisfy your expectations.

---

> > > ### Comment · Reviewer_5YYX · 2025-02-19
> > > **Thanks for the responses**
> > >
> > > Thanks to the authors for their detailed responses. I have read their responses and other reviews. They have addressed my questions. Therefore, I have adjusted the recommendations for the paper.

---

> > > > ### Author Response · Authors · 2025-02-19
> > > > **Follow-Up with the reviewer**
> > > >
> > > > Dear Reviewer,
> > > >
> > > > Thank you for your feedback that helped us to improve our work. We also appreciate your continual engagement and reading our response in a short time given other commitments you have.

---

### Review · Reviewer_s2oY · 2025-02-11

**Summary Of Contributions:**

The paper provides theoretical insights into multi-task learning (MTL) and replay-based continual learning (CL) in overparameterized settings, using linear models as proxies for deep neural networks. It derives explicit formulas for generalization error and knowledge transfer, highlighting the impact of model complexity, task similarity, and dataset size. The study also examines the role of memory buffers in CL and validates the findings through empirical experiments, demonstrating their relevance to deep learning models.

**Audience:**

Yes

**Claims And Evidence:**

Yes

**Requested Changes:**

1. $P_{0|t}$ defined on Page 18 is not a projection. Using the notation $P$ may be misleading; consider changing it or adding a clarifying remark.

2. In Lemma C.6, $P$ should be $X (X^T X)^{-1}X^T$ instead of $X (X^T X)^{-1}X$.

3. In the proof of Lemma C.7 (iii), $P_1X_2$ and $(I - P_1)X_2$ are not independent (e.g., when $P_1X_2 = 0$, then $(I - P_1)X_2 = 0$). The correct statement should be that $P_1X_2$ is uncorrelated with $(I - P_1)X_2$, which is sufficient for applying $E[ab] = E[a]E[b]$. If this is the case, other places, such as the last line of Page 23 ("$P_1X_2$ and $(I - P_1)X_2$ are independent."), should also be revised accordingly.

**Strengths And Weaknesses:**

This is a well-written paper with clearly stated major components. The theoretical insights are interesting and valuable. While most conclusions are derived from a linear model, they provide meaningful intuition and serve as an initial step toward understanding more complex, realistic models. The authors also acknowledge existing work extending linear models, such as neural tangent kernel models.

---

> ### Author Response · Authors · 2025-02-18
> **Responses to Reviewer s2oY**
>
> We appreciate the reviewer's feedback and thank them for highlighting the strengths of our work. Please read our general note first. We have also updated the manuscript with the requested changes, as explained below:
>
>
> 1. That’s a good remark. We added this clarification to the revised version.
>
>
> 2. Good catch! Thanks.
>
>
> 3. Thanks for your question. $P_1 X_2$  and $(I-P_1) X_2$ are indeed uncorrelated as you mentioned. However, notice that for jointly Gaussian distributions, uncorrelatedness is sufficient to imply independence. As a simplified example, let
> $P =
> \\begin{bmatrix}
> \frac{1}{2} & \frac{1}{2}\\\\
> \frac{1}{2} & \frac{1}{2}
> \\end{bmatrix}
> $
> and
> $ X = \\begin{bmatrix}
> x_1\\\\
> x_2
> \\end{bmatrix}
> $, where $x_1, x_2$ are independent samples from $\\mathcal{N}(0,1)$. In this case,
>
> $PX =
> \\begin{bmatrix}
> \frac{1}{2} x_1 + \frac{1}{2} x_2\\\\
> \frac{1}{2} x_1 + \frac{1}{2} x_2
> \\end{bmatrix}
> $ and
> $(I-P)X =
> \\begin{bmatrix}
> \frac{1}{2} x_1 - \frac{1}{2} x_2\\\\
> \frac{1}{2} x_1 - \frac{1}{2} x_2
> \\end{bmatrix}
> $.
>
> Notice that $\frac{1}{2} x_1 + \frac{1}{2} x_2$ and $\frac{1}{2} x_1 - \frac{1}{2} x_2$ are jointly Gaussian (since any linear combination of them is also a Gaussian). They are also uncorrelated and, therefore, independent. Please let us know if this answers your concern.

---

> > ### Comment · Reviewer_s2oY · 2025-02-20
> >
> > Thanks to the authors' detailed explanation. All of my concerns have been addressed.

---

> > > ### Author Response · Authors · 2025-02-20
> > > **Follow-Up with the reviewer**
> > >
> > > Thank you for your time and feedback. We appreciate your review and are glad we could address your concerns.

---

### Review · Reviewer_Nz8y · 2025-02-16

**Summary Of Contributions:**

- Characterize the generalization error of a linear model in a specific multi-task learning (MTL) regression setup.
- Use the analysis to understand the effects of different factors like overparameterization (in a double descent spirit) and task similarity.
- Extend their analysis to a continual learning (CL) setup, analyzing similar factors, but also factors like the size of the replay buffer, which is not fully understood yet in existing CL literature.
- Show empirical results with deeper models, generally agreeing with the analysis on the linear models.

**Audience:**

Yes

**Broader Impact Concerns:**

None.

**Claims And Evidence:**

Yes

**Requested Changes:**

## Major changes
1. The authors use the term "regularization methods" in a way that I find misleading.
Generally, regularization methods solve $\min \left\Vert X_{t}w-y_{t}\right\Vert^2 + \lambda \left\Vert w-w_{t-1}\right\Vert^2_{W}$ where $W$ is some PSD weighting matrix.
Here, the authors use isotropic regularization (which is fine), but also implicitly assume that $\lambda \to 0$. This yields Eq. (13), and is implicitly induced by plain, unregularized SGD [Evron et al. (2022)]. It is of course an interesting setting to investigate (appearing in many recent work), but is *not* what people usually call a "regularization method". The terminology needs fixing.

## Questions
I would like the authors to help me understand the following issues.
1. In the replay Eq. (12), why is the second constraint (on the replay buffer) valid in a noisy setting? Doesn't it heavily depend on the overparameterization level?
1. The authors say on Page 10 "unsharing the classification head is the most effective approach". I'm not sure I understand how this is possible (even technically).
1. On Lemma C.4 and throughout the appendix, the authors use $(X_1^\top X_1)^{-1}$. However, generally this matrix isn't necessarily invertible. Do the authors mean to state their claims w.h.p. or a.s.?
1. How does Term G1 in Eq. (16), when $m=0$, settle with Term G2 in Eq. (10) of Lin et al. (when $T=2,\sigma=0$)? It seems they yield different coefficients.


## Minor changes
1. At the end of page 3, the authors cite Evron et al. (2022) as assuming a noisy data model, while that paper actually assumes a noiseless data model.
1. In the second paragraph on Page 4: should probably use "choose" instead of "chose".
1. Some of the vectors are bold (e.g., $\mathbf{z}_t$, while others are not (e.g., $w_t$).
1. In Figure 2a: It is better to use `semilogx` with the actual x-values rather than manually taking the $log$.
1. The figures would benefit from briefly stating their message (conclusion) at their end.
1. When discussing overparamterization, the authors refer to Goldfarb & Hand (2023). Two recent works [Goldfarb et al. (ICLR 2024), Guha and Lakshman (ICML 2024)] should also be referred to and discussed briefly in that context.
1. Something is wrong in the parenthesis of Eq. (15).
1. Lemma C.4 confused $p_1$ with $P_1$.

**Strengths And Weaknesses:**

## Strengths
- The paper is mostly theoretical but nice experiments were carried out to support the theoretical findings.
- The paper offers a rigorous analysis of MTL in the studied setting, revealing several factors that effect the generalization and transfer.
(I cannot assess how novel this part is.)
- The paper offers one of the first theoretical results on the effect of the replay buffer in CL.
(a similar CL analysis has already been done in [Lin et al. (ICML 2023)]; the novel part here is the replay part.)

## Weaknesses
- The main weakness is that under the investigated data and model assumptions, the analysis becomes uninformative when $p\to\infty$, which is generally the more interesting regime. For instance, Theorem 4.1 simply gives a generalization bound of $1$, while a naive solution of $w=0_d$ would yield the same generalization. This is also apparent from Figure 1.
Moreover, many times the authors discuss phenomena that only occur above that generalization "floor". That is, when the generalization error is already above the naive solution's error.
(The authors seem to have taken a similar analytic approach as a previous CL paper which also suffer from this weakness.).
However, I still find the paper here interesting.

---

> ### Author Response · Authors · 2025-02-18
> **Responses to Reviewer Nz8y**
>
> We sincerely appreciate your thorough feedback and assessment of our work. Please check our general response and find our responses for your specific raised concerns below:
>
> ## Weaknesses
>
> We agree with you that when $p \to \infty$, the analysis becomes less informative. In fact, when $p = \infty$, the loss converges to the $\Vert w^* \Vert ^2$, and this happens due to the fact that the optimal min norm solution becomes the $w=0$ vector. In other words, the learner doesn’t care anymore about the data but just minimizes the norm of the solution as much as possible. Because of this reason, previous studies usually study the behavior as a function of the $\frac{p}{n}$ ratio  (rather than only $p$) as both $n$ and $p$ asymptotically grow to $\infty$ but with a fixed ratio. As we explained in Section 3.2, we ideally want to use MTL only when tasks are collaborative, and $p < \infty$ and in other cases, MTL does not appear to be really effective.
>
> ## Major changes
>
> We understand your concern and agree that the term might be misleading. However, in our context, 'regularization' is drawn from the continual learning (CL) literature. Regularization methods in CL are a well-established category of approaches that are intuitive for readers coming from the CL community. In CL, this term specifically refers to the process where a continual learner acquires new tasks while constraining its deviation from the optimal weights learned in previous tasks. The CL theory papers preceding us used the implicit bias of SGD to implement a regularization-based CL method. But our contribution is to extend this analysis to also include ‘memory-based’ CL methods. Please refer to the revised related work section, where we explained the meaning of regularization in the CL context and please let us know if you believe further clarification is needed.
>
>
> ## Questions
>
> 1. First of all, please note that this approach is the most prevalent practical approach in CL where people store memory samples from previous tasks in an explicit memory buffer and then train a model to fit both the new and old samples stored in the buffer. Using SGD to fit this combined set of training samples will lead to the constrained optimization mentioned in equation 12. Secondly, notice that noise is not crucial in our modeling in the CL section. In fact, both of the theories provided in the main text study the noiseless setting and their conclusions remain unchanged if we set  $\sigma=0$.
>
> 2. In a common MTL setup, it is typically assumed that the task identifier comes with the data. Therefore, we can condition the network architecture over the task id and use a different set of task-specific weights for each task. Please let us know if this explanation answers your question.
>
> 3. When $X$ is a Gaussian matrix with iid entries, $X^T X$ is almost surely invertible (i.e. the probability measure of it not being invertible is zero). Therefore all the expectations we studied work under this assumption. We added a clarification remark reflecting this issue in the revised version.
>
> 4. Although the two terms might look different at first sight, both yield the same coefficients. To see that, notice that we assumed $\Vert w_1^* \Vert = \Vert w_2^* \Vert$ and also their coefficient $r$ is defined as $r = 1 - \frac{n}{p}$.
>
>
> ## Minor changes
>
> Thanks for your detailed feedback. We found them helpful and applied fixes to our revised version.
>
>
> Please let us know if the above answers address your questions or let us know if you have any other concerns.

---

> > ### Comment · Reviewer_Nz8y · 2025-02-21
> >
> > I thank the authors for their response.
> > However, it seems there is still a disagreement on the authors' vast usage of the term "regularization-based" method, which I insist is unconventional and misleading in the current version.
> >
> > As I initially wrote, what the authors currently call "regularization-based" method, is *not* what the CL community considers as regularization methods.
> > 1. Regularization methods are defined in this context as
> > $w_{t+1} = \arg\min_{w} \left(\Vert X_{t+1}^{\top}w - y_{t+1} \Vert^2 + \lambda \Vert w-w_{t}\Vert^2 \right)$ for some *general* $\lambda > 0$
> > [Kirkpatrick et al. (2017); [Li et al. (2023)](https://arxiv.org/abs/2303.10263); [Zhao et al. (2024)](https://proceedings.mlr.press/v235/zhao24n.html)].
> > 1. In Eq. (15), the authors define their CL setup as $w_{t+1} = \arg\min_{w} \Vert w-w_{t}\Vert^2 \text{ s.t. } X_{t+1}^{\top}w= y_{t+1}, \hat{X}^\top w=\hat{y}$ (where $\hat{X},\hat{y}$ is the replay buffer).
> >     1. This is indeed an interesting setup, widely studied in previous work [e.g., Evron et al. (2022); Lin et al. (2023); Goldfarb and Hand (2023); Goldfarb et al. (2024)].
> >     1. However, it is *not* considered a regularization method, but an *unregularized* continual setting.
> >     1. The connection between setups arise, like the authors acknowledged in their response, in unregularized setups trained with SGD, i.e., by the SGD implicit bias (rather than explicit regularization).
> > Specifically with a replay buffer, it will arise when simply minimizing the objective: $\Vert X_{t+1}^{\top}w- y_{t+1}\Vert^2+\Vert \hat{X}^{\top}w-{\hat{y}}\Vert^{2}$ (where $\hat{X},\hat{y}$ is the replay buffer) using SGD, with no constraints and *no* explicit regularization, other than the initialization from the previous task.
> >     1. The studied setup can only be thought of as a *weakly* regularized setup by setting $\lambda \to 0$ in the actual regularized setup, but it is only a very special case.
> >
> > In summary, I still find the current terminology misleading and conflicting with previous CL work.
> > In my humble opinion, this ought to be addressed.

---

> > > ### Author Response · Authors · 2025-02-21
> > > **Follow-Up with the Reviewer**
> > >
> > > Dear Reviewer,
> > >
> > > Thank you for your continual engagement. We value your feedback and really appreciate your time. Given your response, it looks like that your concern is more about terminology which we are hopeful to address. If possible, could you please suggest what might be the best way to resolve this concern? Would it be sufficient to include a discussion in the paper at the beginning to explain the terminology according to what you have explained and then use the term weak regularization in the remainder of the paper?
> > >
> > > Best,
> > >
> > > Our team

---

> > > > ### Author Response · Authors · 2025-02-22
> > > > **Requested Changes Applied**
> > > >
> > > > Dear Reviewer,
> > > >
> > > > We revised the paper as you suggested. We changed the terminology to reflect the fact that we mean "implicit regularization" by what we previously referred to as "regularization", and we hope that the modifications now clearly convey our intention. We kindly ask you to reexamine the revised version, particularly Section 4, and let us know if your concerns are addressed. We gladly look forward to hearing your suggestions.
> > > >
> > > > Best,
> > > > Our team

---

> > > > > ### Comment · Reviewer_Nz8y · 2025-02-24
> > > > > **I am satisfied with the changes**
> > > > >
> > > > > I believe the paper is now clearer and more accurate.
> > > > > I am satisfied with the outcome of the authors-reviewers discussions, and **recommend accepting the paper**.
> > > > > Best of luck.
> > > > >
> > > > > ---
> > > > >
> > > > > P.S - As a last minor comment, in the revised Section 4, around Eq. (14), I think *now* it is appropriate to also cite [Evron et al. (2022)](https://proceedings.mlr.press/v178/evron22a.html), since they were the first to point out this implicit regularization in the CL literature (if I'm not mistaken).

---

> > > > > > ### Author Response · Authors · 2025-02-24
> > > > > > **Follow-up with the reviewer**
> > > > > >
> > > > > > We sincerely appreciate your valuable suggestions and continued engagement in the discussion. Your feedback have been essential in refining our paper.
> > > > > >
> > > > > > Best regards,
> > > > > >
> > > > > > Our team

---

### Author Response · Authors · 2025-02-18
**General Note**

We sincerely appreciate the time and effort each reviewer has taken to provide valuable feedback on our manuscript. Your insights have been instrumental in refining our work and addressing critical concerns. We have carefully considered all the comments and made several revisions to enhance the clarity and robustness of our study. These changes are marked in blue in the revised version of the manuscript. Specifically, we added more detailed explanations regarding the results of prior work and how their results connect with us. We have also incorporated requested changes, such as explicitly defining task similarity, clarifying the implications of our findings at the beginning of the experiments section, and addressing specific technical concerns raised by the reviewers. We believe these revisions significantly improve the clarity of our manuscript and we hope they meet your expectations. Given that the reviewers have found our work sound and the raised concerns are addressable, we hope the reviewers find our responses convincing. We are more than glad to continue engagement with the reviewers if any concerns still remain.

---

### Public Comment · ~Liangzu_Peng1 · 2025-05-12
**Nice work, and some comments on prior work**

Hello,

Congrats on the publication. I was reading the paper with great interest and came to the openreview page to extract bib info.

After reading the metareview, I wanted to add two references related to the claim of a reviewer and to a problem formulation of the paper:

- Theorem 3 of my paper (https://arxiv.org/abs/2305.00316) is an earlier result on experience replay. (That theorem has a very simple proof and is not optimal in many ways.)
- The formulation considered in Eq 16 (Implicit Regularization+Replay-Based Continual Learning) is almost identical to the affine projection algorithm that is proposed in 1975 and then 1984 and that originates from the adaptive filtering literature. There are two differences though: (1) the memory buffer is chosen differently, (2) the affine projection algorithm considers one sample at a time (or one sample per task), while the paper considers multiple samples per task. Theoretical guarantees for the affine projection algorithm have existed for a while; e.g., see the book (Adaptive Filters, 2008).

A review of the affine projection algorithm can be found in my recent tutorial (https://arxiv.org/abs/2504.17963) or in the corresponding sections of the book (Machine Learning: A Bayesian and Optimization Perspective).

I wish that helps.

---

> ### Author Response · Authors · 2025-05-16
>
> Hi Liangzu,
>
> Thanks for reaching out and also for bringing these works to our attention. I appreciate you taking the time to engage with our paper. I just had a chance to quickly skim through the papers you shared. I acknowledge their connection to our results, but I'd like to emphasize the distinction as well. For example, the bound in Theorem 3 of ICL [1] is based on statistical learning analysis, and becomes quickly loose, especially when the learner class is complex, but of course with the advantage of being agnostic to the data distribution (as opposed to our work, which assumes Gaussian features).
>
> Compared to the APA guarantees, as you pointed out, their setup considers a single sample per time step, and they have no notion of "task", an assumption which does not hold in the current CL framework. Our theories are built on the random matrix theory, and it enables us to go beyond single-sample setups. Moreover, as far as I understood from your tutorial, their guarantees work only under the full memory setup, i.e., when all samples are explicitly stored in a memory buffer. Again, something which is far from realistic CL models.
>
> I enjoyed reading through your tutorial, and thanks again for bringing it to our attention.
> We'd like to hear more from you, and please stay connected.
>
> [1] https://arxiv.org/abs/2305.00316.
> [2] https://arxiv.org/abs/2504.17963

---

### Decision · Action_Editor_fHtW · 2025-03-12

**Recommendation:** Accept as is

**Comment:**

This work makes a solid contribution that, among other things, shows theoretical results exhibiting the effect of replay buffer size in continual learning. The experiments are interesting. The authors show that a double descent phenomenon occurs in the continual learning setting. This work is a welcome contribution to TMLR.

**Audience:**

There is a large audience for works related to continual learning, and there is a subset of this audience that will appreciate theoretical results in this area. To quote one reviewer from the post-review discussion, "The paper offers one of the first theoretical results on the effect of the replay buffer in CL.". So, for sure, there is an audience for this work.

**Claims And Evidence:**

The reviewers found the results in this paper to be rigorous and also appreciated the experiments. On the theory side, it was not clear to what extent the authors are using new techniques or how these results quantifiably compare to previous results. Even so, there is no doubt about the authors' theoretical and experimental support for their results.